# Crosstalk between Autophagy and RLR Signaling

**DOI:** 10.3390/cells12060956

**Published:** 2023-03-21

**Authors:** Po-Yuan Ke

**Affiliations:** 1Department of Biochemistry & Molecular Biology, Graduate Institute of Biomedical Sciences, College of Medicine, Chang Gung University, Taoyuan 33302, Taiwan; pyke0324@mail.cgu.edu.tw; Tel.: +886-3-211-8800 (ext. 5115); Fax: +886-3-211-8700; 2Liver Research Center, Chang Gung Memorial Hospital, Taoyuan 33305, Taiwan

**Keywords:** autophagy, RLR, selective autophagy, antiviral response, innate immunity

## Abstract

Autophagy plays a homeostatic role in regulating cellular metabolism by degrading unwanted intracellular materials and acts as a host defense mechanism by eliminating infecting pathogens, such as viruses. Upon viral infection, host cells often activate retinoic acid-inducible gene I (RIG-I)-like receptor (RLR) signaling to induce the transcription of type I interferons, thus establishing the first line of the innate antiviral response. In recent years, numerous studies have shown that virus-mediated autophagy activation may benefit viral replication through different actions on host cellular processes, including the modulation of RLR-mediated innate immunity. Here, an overview of the functional molecules and regulatory mechanism of the RLR antiviral immune response as well as autophagy is presented. Moreover, a summary of the current knowledge on the biological role of autophagy in regulating RLR antiviral signaling is provided. The molecular mechanisms underlying the crosstalk between autophagy and RLR innate immunity are also discussed.

## 1. Introduction

Autophagy is a fundamentally catabolic process that degrades unnecessary intracellular materials to support the recycling of nutrients, thus promoting the regeneration of energy sources and balancing metabolic homeostasis. In addition to basal autophagy, a variety of stimuli, including deprivation of nutrients, aggregation and/or unfolding of proteins, damage to organelles, and infection by pathogens, can specifically activate autophagy to resolve cellular stresses. Deregulation of autophagy is involved in the pathogenesis of human diseases, such as cancer, neurodegenerative diseases, infectious diseases, and metabolic disorders. Several viral infections may induce autophagy to reconstitute the membranous compartments for virus replication and to repress the innate antiviral response, which benefits virus replication. In contrast, virus-triggered autophagy may be used to restrict viral growth by eliminating the infecting virus, acting as a host defense mechanism against viral infection. Upon virus infection, pattern recognition receptors (PRRs), such as retinoic acid-inducible I (RIG-I)-like receptors (RLRs), recognize pathogen-associated molecular patterns (PAMPs) and subsequently trigger a downstream signal cascade to induce the production of interferons (IFNs) and cytokines, thus stimulating innate antiviral immunity. However, to achieve successful replication, viruses have evoked strategies to repress the RLR-triggered antiviral response and may also interfere with autophagic degradation. In recent years, numerous studies have provided insights into the molecular interactions between virus-activated autophagy and RLR innate immune signaling. In this review, we first summarize the current understanding of the molecular regulation of RLR signaling and autophagy. Then, we provide an overview of the physiological importance and molecular mechanism by which autophagy regulates the antiviral response of RLRs.

## 2. RLR Antiviral Signaling

### 2.1. Functional Molecules Involved in RLR Signaling

During viral infection, the binding of virus genome-containing PAMPs by PRRs triggers a signaling cascade that transcriptionally activates the gene expression of IFNs and cytokines in infected cells, ultimately promoting the synthesis and secretion of IFNs and cytokines [1,2,3,4]. PRRs are sensing molecules localized in the cytosol and membranous compartments within cells and are responsible for recognizing viral-derived PAMPs and stress cell-produced danger-associated molecular patterns (DAMPs) [5,6,7,8,9]. To date, three classes of PRRs have been shown to activate the innate immune response, including the membrane-bound Toll-like receptors (TLRs) responsible for sensing viral PAMPs in endosomes [7,8,9,10,11,12,13], cytosolic RLRs responsible for sensing viral RNA [6,7,8,9,14,15], and cyclic GMP-AMP (cGAMP) synthases (cGASs) responsible for sensing DNA in the cytosol [7,8,9,16,17]. In addition to these PRRs, the cytosolic NBD leucine-rich repeat-containing receptor (NLR) family, such as absent in melanoma 2 (AIM2)—an initiator of the assembly of inflammasome complex in immune cells—contributes to PAMP and DAMP sensing and leads to proteolytic activation of caspases, thereby activating immune responses and inflammation [18,19,20].

Three RLRs that possess DExD/H box helicase activity, RIG-I [21], melanoma differentiation association gene 5 (MDA5) [22], and laboratory of genetics and physiology 2 (LGP2) [23], were identified in the 2000s. All three RLRs contain an ATP-dependent DExD/H box-containing RNA helicase domain for the recognition of double-stranded RNA (dsRNA) in the central region and a C-terminal repressor domain (CTD) for binding to RNA- and Zinc ligands (Figure 1A) [22,24,25,26,27,28,29]. A pincer-link domain connects the C-terminus of the helicase domain (HELICc) and the CTD (Figure 1A) [22,24,25]. Additionally, RIG-I and MDA5 contain two caspase activation and recruitment domains (CARDs) at the N-terminus, which transduce signaling to downstream effectors to induce the IFN response [22,24,26,27,30]. Moreover, these tandem CARDs of RIG-I interact with the helical insertion segment of the helicase domain in cells under steady-state conditions, forming an autorepressor conformation which interferes with the binding of ligand RNA to RIG-I [26,27,28,29,31]. Upon viral infection, binding of the RNA ligand to the CTD of RIG-I induces an ATP-dependent conformational change in RIG-I and relieves the N-terminal CARDs, which associates with the downstream signaling molecule [26,27,28,29,31,32]. In contrast, LGP2 lacks these N-terminal CARDs, but it retains the core helicase domain that can bind to dsRNA and has been shown to regulate the biological functions of RIG-I and MDA-5 (Figure 1A) [25,33,34].

In addition to RIG-I and MDA5, mitochondrial antiviral signaling (MAVS, also known as IFN-beta promoter stimulator-1 [IPS-1], virus-induced signaling adaptor [VISA], and CARD adaptor inducing IFN-β [Cardif]) also contains a CARD at its N-terminal regions (Figure 1A), which can interact with the CARDs of RIG-I and MDA5 and a transmembrane domain at the C-terminus for localization on the outer membrane of mitochondria (MOM) (Figure 1B) [35,36,37,38]. The association between the CARDs of RIG-I/MDA5 and MAVS activates downstream signaling. Moreover, oligomerization of MAVS through N-terminal CARDs assembles into protease- and detergent-resistant prion-like structures on the MOM and promotes the recruitment of the ubiquitin E3 ligases tumor necrosis factor (TNF) receptor-associated factor 2 (TRAF2), TRAF5, and TRAF6 [39,40]. Subsequently, these ubiquitin E3 ligases produce the polyubiquitin chain that recruits the nuclear factor kappa-light-chain-enhancer of activated B (NF-kB) essential modulator (NEMO), which in turn activates TANK binding kinase 1 (TBK1) and inhibitor of nuclear factor kappa B (Ik-B) kinase (IKK) to phosphorylate interferon response factor 3 (IRF3) and Ik-B, respectively (Figure 1B) [39,40]. The homodimers of phosphorylated IRF3 and NF-kB then translocate into the nucleus and transactivate the gene expression of IFN and proinflammatory cytokines, respectively (Figure 1B) [39,40]. Then, type I IFNs, such as IFN-α and IFN-β, are expressed and secreted in virus-infected cells and bind to the IFN-α receptor (IFNAR), which consists of IFNAR1 and IFNAR2 subunits [1,2,3,41].

The binding of IFNs to IFNAR activates the IFNAR-associated receptor tyrosine kinases tyrosine kinase 2 (TYK2) and Janus kinase 1 (JAK1) to phosphorylate signal transducer and activator of transcription 1 (STAT1) and STAT2. Subsequently, the tyrosine phosphorylation of STAT1 and STAT2 induces the formation of homodimers and triggers translocation into the nucleus. Before entering the nucleus, this STAT1-STAT2 homodimer associates with interferon response factor 9 (IRF9), forming a heterotrimeric protein complex, IFN-stimulated gene factor 3 (ISGF3). ISGF3 is able to bind to IFN-stimulated response elements (containing the consensus sequence TTTCNNTTTC) and thus transcriptionally activate the gene expression of IFN-stimulated genes (ISGs). Several types of ISGs have been identified, and their biological functions, including antiviral activity, are less understood, although some of them have been shown to repress virus entry and replication [1,2,42,43].

### 2.2. Recognition of Viral RNA by RLRs

RLRs are ubiquitously expressed in most tissues and are mostly dominantly expressed in the cytosol [6,14,15]. RIG-I has been shown to recognize dsRNA or single-stranded RNA (ssRNA) with triphosphate or diphosphate moieties at the 5′-end and is approximately 10–300 bp in length [44,45,46,47]. Additionally, the poly-uridine (U)/U-core (UC) (poly-U/UC) located within the 3′-untranslated region of the hepatitis C virus (HCV) RNA genome (3′-UTR) [48,49,50] and the adenine-uridine (AU)-rich sequence in the viral RNA of measles virus [51,52] serve as additional signatures for determining the specificity of viral PAMP binding to RIG-I. Moreover, the cleavage of viral RNA by RNase L promotes the release of small PAMP RNA containing a 5′-triphosphate (5′-ppp) moiety to induce the RIG-I-mediated antiviral IFN response [53,54]. In contrast to RIG-I, MDA5 binds to longer dsRNA sequences (approximately 1–2 kilobase [Kb] pairs), which presumably are replication intermediates of viral RNA that form a high-order RNA structure [55,56,57,58]. Genetic knockout studies in mice showed that RIG-I and MDA5 play distinct role(s) in the recognition of different RNA viruses [59,60]. To date, RIG-I has been extensively shown to recognize several human pathogenic viruses such as influenza A virus (IAV) [46,61,62,63,64], HCV [48,49,50,54], dengue virus (DENV) [65,66,67], Zika virus (ZIKV) [65,68], and West Nile virus (WNV) [69,70,71], and activate antiviral immunity in infected human cells. Similarly, several viruses that can lead to human diseases, including picornavirus [55,72], HCV [73,74], ZIKV [68], WNV [71], and severe acute respiratory syndrome coronavirus 2 (SARS-CoV-2) [75,76], also trigger antiviral IFN responses through MDA5.

### 2.3. Post-Translational Modification(s) of RLR and Regulation of RLR Signaling

Several post-translational modifications (PTMs) have been found to extensively modulate RLR-triggered antiviral immunity, such as phosphorylation, acetylation, sumoylation, deamination, ISGylation, and ubiquitination [6,7,8,9,14,15]. Conformational changes, including oligomerization and the formation of highly ordered and/or filamentous structures of RIG-I, MDA5, and MAVS, have been shown to regulate the activation of the RLR antiviral IFN response [6,7,8,9,14,15]. Protein kinase C (PKC)-α and PKC-β-induced protein phosphorylation at serine (Ser) residue 8 and threonine (Thr) residue 170 within the CARD was shown to repress the signal transduction of RIG-I antiviral immunity by interrupting the binding of RIG-I to tripartite motif (TRIM) 25 (TRIM25), a ubiquitin ligase required to positively regulate RIG-I downstream IFN signaling (Figure 2) [77,78,79]. Additionally, phosphorylation at Thr770, Ser854, and Ser855 within the CTD of RIG-I by casein kinase II (CK-II) was found to be necessary for maintaining the autorepressor conformation of RIG-I (Figure 2) [80], thereby preventing the improper activation of RIG-I antiviral immunity in the absence of viral infection. In addition, the induction of RIG-I signaling promotes the activation of death-associated protein kinase 1 (DAPK1) to phosphorylate Thr667 within the helicase domain and interfere with RIG-I recognition of 5′-triphosphate-dsRNA, acting as feedback inhibition of the RIG-I antiviral response (Figure 2) [81]. Similarly, MDA5 is also negatively regulated by the phosphorylation of Ser88 on the CARD and Ser828 on the CTD of MDA5 (Figure 2) [82,83]. In contrast, dephosphorylation of RIG-I at Ser8 and Thr170 and MDA5 at Ser88 and Ser828 by protein phosphatase 1 (PP1)-α and PP1-γ was demonstrated to trigger RIG-I and MDA5-mediated antiviral immunity (Figure 2) [83,84].

Acetylation of RIG-I at lysine (Lys) residue 858 (Lys858) and Lys909 within the CTD domain was shown to interrupt the oligomer formation of RIG-I, which is necessary to activate RIG-I, thus repressing antiviral immunity (Figure 2) [85,86]. In contrast, histone deacetylase 6 (HDAC6)-mediated deacetylation of RIG-I reversely triggers the sensing of viral RNAs by RIG-I and activates downstream antiviral signaling (Figure 2) [85,86]. The sumoylation of RIG-I at Lys96 and Lys888 and MDA5 at Lys43 and Lys865 was demonstrated to inhibit the degradation of RIG-I and MDA5 in the cell resting state and promote the PP1-mediated dephosphorylation of RIG-I and MDA5 during viral infection (Figure 2) [87]. Conversely, sentrin/SUMO-specific protease 2 (SENP2)-mediated desumoylation of RIG-I and MDA5 led to Lys48 polyubiquitination and proteasomal degradation at the late stage of viral infection (Figure 2) [87]. In addition, the conjugation of IFN-stimulated gene 15 (ISG15) to RIG-I by ISGylation and deamination of RIG-I represent additional regulatory mechanisms for activating RIG-I antiviral immunity [88,89]. On the other hand, adenosine deaminase acting on RNA 1 (ADAR1), an ISG with RNA editing, was shown to restrict the recognition of viral RNAs by RLRs and repress the downstream antiviral immune response [90,91,92].

Ubiquitination has been widely demonstrated to play multiple roles in regulating RLR signaling. Linkage of the Lys-63-linked polyubiquitin chain onto Lys172 within the N-terminal CARDs of RIG-I by TRIM25 was first reported to serve as a prerequisite signal for binding to MAVS (Figure 2), thus controlling the induction of the downstream antiviral IFN response [93]. In addition to TRIM25, another ubiquitin E3 ligase, RING finger protein 135 (RNF135, also named Riplet and REUL), induces Lys-63-linked polyubiquitination of RIG-I CARDs (Lys154, Lys164, and Lys172) and CTD (Lys788, Lys849, and Lys851) to activate the antiviral IFN response (Figure 2) [94,95,96,97]. Moreover, mex-3 RNA binding family member C (MEX3C, also known as RNF194) also triggers the linkage of Lys-63-linked polyubiquitin to Lys45, Lys99, and Lys169 within the CARDs of RIG-I to promote the immune response against viral infection (Figure 2) [98]. In contrast to the positive role of Lys-63-linked polyubiquitination, the linkage of the Lys-48-linked polyubiquitin chain to RIG-I promotes the degradation of RIG-I, thereby negatively regulating the activation of the antiviral IFN response [99,100,101]. To date, at least three ubiquitin ligases, including RNF122, RNF125, and casitas B-lineage lymphoma proto-oncogene (Cbl), have been indicated to promote Lys-48-linked polyubiquitination at the Lys115 and Lys146 residues of RIG-I CARDs and Lys813 on the CTD of RIG-I, thus destabilizing RIG-I (Figure 2) [99,100,101]. In contrast, deubiquitination (DUB) enzymes, such as ubiquitin-specific protease (USP) 4 (USP4) and USP15, enhance RLR antiviral immunity by removing the Lys48-linked polyubiquitination of RIG-I (Figure 2) [102,103]. Other DUB enzymes, USP3 and CYLD lysine 63 deubiquitinase (CYLD), bind to CARDs of RIG-I and cleave Lys-63-linked polyubiquitin to repress the RIG-I-mediated IFN response (Figure 2) [104,105]. Analogously, TRIM65-mediated Lys-63-linked polyubiquitin at Lys743 within the helicase domain of MDA5 was shown to pivotally activate MDA5 antiviral signaling [106], whereas RNF125 represses MDA5 antiviral function by promoting Lys-48-linked polyubiquitination (Figure 2) [99]. USP17 has been identified as a DUB enzyme involved in the positive regulation of the IFN response by deubiquitinating MDA5 as well as RIG-I [107]. Very recently, ovarian tumor protease (OTU) deubiquitinating enzyme 3 (OTUD3), a deubiquitination enzyme, was reported to bind to MDA5 and RIG-I and ultimately remove their Lys-63-linked polyubiquitination to diminish the signal transduction of RLRs, thus inhibiting the antiviral immune response against RNA viruses (Figure 2) [108]. Apart from its function of complexing with Lys on RLRs, the unanchored Lys-63-linked polyubiquitin chain also serves as a ligand of RLR CARDs and potentiates RLR antiviral signaling [109,110].

In the past decade, mounting lines of evidence have demonstrated that the assembly of filamentous and highly ordered structures of RLRs, including RIG-I and MDA5, critically activates RLR antiviral signaling by enhancing their binding affinity to dsRNA and MAVS [110,111,112,113,114,115,116,117,118]. In the presence of an unanchored Lys-63-linked polyubiquitin chain and ATP hydrolysis, the CARDs of RIG-I assemble on RNA and form a tightly bound tetramer, which forms single-competent aggregates with the CARDs of MAVS [110,111,112,118]. In contrast to RIG-I, the head-to-tail assembly of MDA5 forms a filament along the long dsRNA sequence, allowing MDA5 to discriminate between self- and nonself RNA via assembly and disassembly dynamics [113,114,115,116]. It is also noted that ATP hydrolysis is required to disassemble MDA5 filaments rather than to oligomerize MDA5 [113,114,115,116]. In addition to RLRs, viral infection induces the aggregation of MAVS on the MOM and forms prion-like fibers that are resistant to detergent and proteases [39,40]. MAVS aggregates on the MOM and activates RLR antiviral signaling not only by associating with the RIG-I tetramer to form single-competent aggregates [110,111,112,118] but also by recruiting ubiquitin E3 ligases to produce polyubiquitin chains [39,40]. In addition to mitochondria, MAVS localized on peroxisomes and mitochondria-associated endoplasmic reticulum (ER) membranes (MAMs) also serves as a platform for innate antiviral immunity [119,120,121]. In particular, viral infection induces the assembly of a RIG-I translocon consisting of RIG-I, TRIM25, and 14-3-3ε on MAMs to activate the RLR antiviral response by promoting RIG-I polyubiquitination and interaction with MAVS [121].

### 2.4. Gene Polymorphism(s) of RLR and Modulation of RLR Antiviral Signaling

Single nucleotide polymorphisms (SNPs) in RLR have been identified and shown to regulate IFN response and affect viral susceptibility. The mutation of serine at residue 183 of RIG-I to isoleucine (Ser183Ile) has been shown to inhibit the polyinosinic acid-polycytidylic acid (poly ([I:C]), 5′-triphosphate-RNA, Sendai virus (SeV), and IAV-induced activation of IFN-β promoter, presumably increasing the formation of RIG-I self association [122,123]. In addition, the mutation of arginine at cysteine (Arg7Cys) also reportedly decreases the activation of the IFN-β promoter in IAV-infected cells [124]. In contrast, a nonsense mutation of glutamic acid at residue 627 (E627) leads to deletion of the C-terminal region and dsRNA binding activity of MDA5, which represses the poly (I:C)-triggered activation of the IFN-β promoter [123]. In addition, an inherited missense mutation in MDA5 (mutation of lysine at residue 365 to glutamic acid, K365E) also reportedly represses the poly (I:C)-induced activation of the IFN-β promoter and the IFN-driven ISRE activation of the ISRE promoter [125]. Additionally, K365E mutant MDA5 increases the replication of human rhinovirus (HRV) in patient-derived nasal epithelial cells [125]. Type 1 diabetes (T1D)-associated variants in MDA5, including mutations of alanine at residue 946 to threonine (A946T), isoleucine at residue 923 to valine (I923V), and glutamic acid at residue 627 to the stop codon (E627X), have also been demonstrated to decrease the rotavirus (RV)-induced upregulation of IFN-β mRNA and increase the replication of RV in infected cells [126]. These studies collectively indicate that SNPs and germline mutations of RLR may disturb innate antiviral immunity and enhance susceptibility to virus infection.

## 3. Autophagy

### 3.1. Types of Autophagy and Autophagy-Related Molecules

Autophagy is a self-digestion process that recycles nutrients and regenerates energy through lysosomal degradation of unwanted intracellular components, such as damaged organelles and aggregated/unfolded proteins [127,128,129]. Proper and precise control of autophagy primarily maintains cellular homeostasis, whereas dysregulation of autophagy has been widely shown to be associated with human diseases [129,130,131]. Three types of autophagy have been identified in eukaryotes, including microautophagy, macroautophagy, and chaperone-mediated autophagy (CMA); all three kinds of autophagy involve lysosomal degradation (Figure 3) [129,132,133,134]. Microautophagy involves the protrusion and/or invagination of the lysosomal membrane to sequester the components of the cytosol, including organelles, and deliver them to the lysosomal lumen for degradation (Figure 3A) [135,136,137,138]. CMA is also a lysosome-dependent degradation process that selectively eliminates substrates that possess the pentapeptide “Lys-Phe-Glu-Arg-Gln” (KFERQ) motif (Figure 3B) [139,140,141]. After recognition by heat shock cognate 70 kDa protein (HSC70), which specifically binds to lysosomal membrane protein 2A (LAMP2A) on the lysosomal membrane, CMA cargoes are then transported into the lumen of lysosomes and degraded by lysosomal proteases (Figure 3B) [139,140,141]. Macroautophagy, the most common form of autophagy, is a stepwise vacuole biogenesis process involving the formation of double-membranous autophagosomes that sequester cytoplasmic portions and ultimately deliver them to lysosomes for decomposition (Figure 3C) [129,134,142,143]. The completion of autophagy relies on the coordinated regulation of functional autophagy-related genes (ATGs) and the successful rearrangement of intracellular membranes, which is initiated by the nucleation of the isolation membrane (IM)/phagophore [129,142,143,144,145]. When cells are starved of nutrients, suppression of the serine/threonine protein kinase mammalian target of rapamycin (mTOR) leads to translocation of the unc-51-like kinase (ULK) complex containing ULK1/2, ATG13, ATG101, and RB1-inducible coiled-coil 1 (RB1CC1, otherwise known as FIP200) from the cytoplasm to a unique compartment derived from the ER membrane [146,147,148,149]. Then, the ULK complex activates the class III phosphatidylinositol-3-OH kinase (PI(3)K) complex (containing Vps34/PI3KC3, ATG14, Beclin 1, and Vps15) on the ER-reconstituted microdomain, leading to the biosynthesis of phosphatidylinositol-3-phosphate (PtdIns(3)P) [145,150,151,152]. The newly generated PtdIns(3)P subsequently induces the relocalization of double-FYVE-containing protein 1 (DFCP1) and WD-repeat domain PtdIns(3)P-interacting (WIPI, also known as ATG18) to induce the emergence of an ER-reconstituted and omegasome-like IM/phagophore [145,150,151,153,154]. Furthermore, the ER-associated vacuole membrane protein 1 (VMP1) promotes crosstalk between the ER and IM/phagophore and the recruitment of the class III PI(3)K complex by direct binding to Beclin-1 [155,156,157]. Moreover, the multi-spanning membrane protein ATG9a triggers membrane movement and lipid mobilization from other organelles, including the trans-Golgi network (TGN), plasma membrane, and lipid droplet to ER subdomain, thus supporting the formation of IM/phagophore [158,159,160,161,162]. In addition to the ER [163,164], various types of intracellular membranous compartments, including mitochondria [165], Golgi apparatus [166], recycling endosome [167,168], MAMs [169], and plasma membrane [170], also provide the required membranes for the emergence of IM/phagophore.

When IM/phagophores are nucleated on the preautophagosomal (PAS) structure, two ubiquitin-like (UBL) conjugation systems subsequently facilitate the expansion and enclosure of IM/phagophores into double membranous autophagosomes (Figure 3C) [149,171,172,173,174,175]. The enzymatic reaction composed of ubiquitination activating enzyme 1 (E1), ATG7, and ubiquitin-conjugating enzyme 2 (E2), ATG10, induces the covalent conjugation of the UBL molecule ATG12 to ATG5, forming the ATG12-ATG5 conjugate. Then, the ATG12-ATG5 conjugate further interacts with ATG16L to form an ATG12-ATG5-ATG16L heterotrimeric complex [171,172,176,177,178]. Moreover, the conjugation of phosphatidylethanolamine (PE) to ATG8/LC3 family proteins (so-called lipidation of ATG8/LC3) requires another UBL conjugation machinery involving ATG7 (E1) and ATG3 (E2) [179,180,181]. The C-termini of ATG8/LC3 proteins are first cleaved by ATG4 family proteases, producing ATG8/LC3-I, which is then conjugated to PE to generate ATG8/LC3-II, namely, lipidated ATG8/LC3 (also known as ATG8/LC3-PE) [179,180,181]. Then, ATG8/LC3-II promotes the expansion of the PAS membrane and membrane fusion to facilitate the maturation of autophagosomes [174,182,183], whereas ATG12-ATG5-ATG16L (presumably functions as an E3 ligase) facilitates the conjugation of PE to ATG8/LC3-I [184,185]. Despite these UBL conjugation systems, UV radiation resistance-associated gene (UVRAG) protein interacts with the class III (PI(3)K) complex and participates in the trafficking of endosomes and autophagosome formation [151,186,187]. Mature autophagosomes subsequently fuse with lysosomes (Figure 3C), forming autolysosomes in which acidic lysosomal proteases degrade the sequestrated components to recycle nutrients and regenerate energy. Several classes of proteins have been shown to function in autophagosome–lysosome fusion [175,188,189,190,191,192]. The Rab-interacting lysosomal protein (RILP) and FYVE and coiled-coil domain-containing 1 (FYCO1), two regulators involved in microtubule dynamics, have been shown to interact with the autophagosome-bound small GTPase Ras-related protein 7 (Rab7), thereby promoting the movement of autophagosomes along with microtubules [193,194,195,196,197]. In addition, actin remodeling by HDAC6 also facilitates the maturation of autolysosomes [198]. In addition to autophagosome transport through the cytoskeleton, pleckstrin homology domain-containing protein family member 1 (PLEKHM1) and homotypic fusion and protein sorting (HOPS) complex associated with syntaxin 17 (STX17) promote the fusion of autophagosomes with lysosomes [199,200,201]. Moreover, the protein complex composed of ATG14L, vesicle-associated membrane protein 8 (VAMP8), synaptosome-associated protein 29 (SNAP29), and STX17 stimulates the tethering of membranes involved in autophagosome–lysosome fusion [202,203]. Furthermore, ATG8/LC3 family proteins play pivotal roles in the fusion of autophagosomes with lysosomes by recruiting PLEKHM1 [204]. When the autophagy process is completed, the reactivation of mTOR by nutrient recycling subsequently terminates autophagy by coordinately suppressing the initiation of autophagy and activating autophagic lysosome reformation (ALR) [205]. Several molecules, including phospholipid PtdIns(4,5)P(2), the functional molecules involved in endocytosis, clathrin, and motor movement, kinesin, the lysosomal efflux permease, and the ubiquitin E3 ligase Cullin 3-Kelch-like protein 20 (KLHL20), have been shown to participate in ALR activation at the termination stage of autophagy [206,207,208,209,210,211,212,213,214,215,216].

### 3.2. Selective Autophagy

Although autophagy is known as a non-specific degradation process, accumulating lines of evidence indicate that autophagy may selectively sequester intracellular components and deliver them to lysosomes for degradation, a process called “selective autophagy” (Figure 3C) [217,218,219,220,221]. Through recognition of protein ubiquitination and/or association with adaptor proteins and the specific interaction between the autophagosomal ATG8/LC3 family proteins, the cargo receptors of selective autophagy can specifically target proteins and organelles for autophagic degradation (Figure 3C) [220,221,222,223,224,225,226,227,228]. Several types of cargo receptors, such as the neighbor of BRCA1 (NBR1), calcium-binding and coiled-coil domain-containing protein 2 (Calcoco2, also named NDP52), p62/sequestosome 1 (p62/SQSTM1), optineurin (OPTN), BCL2/adenovirus E1B 19 kDa protein-interacting protein 3 (BNIP3)-like (BNIP3L/Nix), and Tax1-binding protein 1 (Tax1 bp1), have been identified to function in selective autophagy [220,221,224,225,227,228]. Most of these cargo receptors contain the ubiquitin-associated domain (UBA) that is responsible for the recognition of ubiquitinated proteins and the LC3-interacting regions (LIRs), which are required for ATG8/LC3 family protein binding [220,221,224,225,227,228]. In addition to LIRs located within cargo receptors, two unique interacting domains for binding to ATG8/LC3 family proteins, including ATG8-interacting motifs (AIMs) and GABARAP-interacting motifs (GIMs) within ATGs and adaptor proteins, have been shown to participate in the sequestration of cargoes within selective autophagy and the local biogenesis of autophagosomes [229,230,231,232]. Specific elimination of organelles, including mitochondria, peroxisomes, lysosomes, ER, ribosomes, lipid droplets (LDs), and nuclei, by selective autophagy (referred to as “organellophagy”) not only maintains the integrity of organelles but also promotes the regeneration of them (Figure 3C) [217,219,233].

Mitophagy is a specific kind of selective autophagy that uniquely targets mitochondria for degradation [234,235]. The fission and depolarization of mitochondria induced by mitochondrial damage inhibit the proper proteolysis of PTEN-induced putative kinase 1 (PINK1) by presenilin-associated rhomboid-like protein (PARL) within mitochondria, leading to stabilization of PINK1 on the MOM [236,237]. MOM-bound PINK1 induces the recruitment of Parkin, a ubiquitin E3 ligase, and phosphorylates serine 65 residues within Parkin and ubiquitin [238,239,240,241,242,243]. Parkin in turn promotes the ubiquitination of mitochondrial proteins [238,239,240,241,244], thereby inducing the recruitment of selective autophagy cargo receptors, such as p62/SQSTM1, NDP52/Calcoco2, and OPTN, for the removal of damaged mitochondria in a PINK1/Parkin-dependent manner (Figure 3C) [235,245]. The TBK1-induced phosphorylation of Ser177, Ser473, and Ser513 within OPTN and Ser403 within p62/SQSTM1 facilitates the recruitment of cargo receptors for mitophagy [246,247,248]. In addition to these cargo receptors, Toll-interacting protein (Tollip) and the mitochondrial adenine nucleotide translocator (ANT) complex serve as other mitophagy receptors for PINK1/Parkin-dependent mitophagy (Figure 3C) [249,250,251,252,253,254]. To facilitate the sequestration of ubiquitinated mitochondria by autophagy, DFCP1 and WIPI/ATG18 family proteins, two regulators of initiating autophagy, are coincidently recruited to facilitate the emergence of IM/phagophore proximal to damaged mitochondria [245]. Moreover, the spatiotemporal translocations of FIP200 and ATG13 by TBK1 phosphorylation and OPTN [255] and the omegasome-proximal PtdIns(4,5)P_2_ and F-actin organization [256] also promote the constitution of an omegasome-like structure and the disassembly of mitoaggregates to facilitate the removal of damaged mitochondria during mitophagy. However, deubiquitination of MOM proteins by DUB enzymes USP30 and USP35 suppresses mitophagy [257,258]. In contrast, various mitochondrial proteins, including BNIP3, FUN14 domain-containing 1 (FUNDC1), BNIP3L, yeast ATG32, prohibitin 2 (PHB2), and nitrophenylphosphatase domain and nonneuronal SNAP25-like protein homolog (NIPSNAP) family proteins, activate mitophagy by interacting with ATG8/LC3 family proteins (Figure 3C) [259,260,261,262,263,264,265,266].

To remove the damaged ER by organellophagy (referred to as ER-phagy) [267,268,269], ATG39, ATG11, and ATG40 in yeast [270] and the family with sequence similarity 134, member B (FAM134B), and reticulon family proteins in mammals [271,272,273] serve as cargo receptors for targeting the stressed ER for degradation (Figure 3C). In addition, testis-expressed sequence 264 protein (TEX264) [274,275], Atlastins 3 (ATL3) [276,277], cell-cycle progression gene 1 (CCPG1) [278], translocon, SEC62 [279], and progesterone receptor membrane component 1 (PGRMC1) [280] act as cargo receptors of ER-phagy (Figure 3C). Moreover, the elimination of harmful peroxisomes by organellophagy, so-called pexophagy, relies on several identified cargo receptors, including yeast ATG36 [281,282,283], mammalian NBR1 [284], and mammalian p62/SQSTM1 (Figure 3C) [285]. Additionally, two kinases, yeast Hrr25 and mammalian ataxia-telangiectasia-mutated (ATM), induce the phosphorylation of ATG36 and p62/SQSTM1, respectively, to positively regulate pexophagy [281,285,286]. In addition to phosphorylation, polyubiquitination of several peroxisomal (PEX) membrane proteins, such as PEX5 and the 70-kDa PEX membrane protein (*PMP70*) by the ubiquitin E3 ligase PEX2, also promotes NBR1-dependent pexophagy in mammals [287]. Very recently, the induced translocation of a ubiquitin E3 ligase, STIP1 Homology and U-Box Containing Protein 1 (Stub1), was shown to promote the turnover of oxidized peroxisomes through ubiquitination-dependent pexophagy (Figure 3C) [288]. Organellophagy is also involved in the turnover of other organelles and proteins, including lysosomes (lysophagy) [289,290,291], nuclei (nucleophagy) [270,292], ribosomes (ribophagy) [293,294,295], protein aggregates (aggrephagy) [198,296,297,298,299], LDs (lipophagy) [300,301,302,303], and ferritin (ferritinophagy) [304,305,306,307], through specific cognate cargo receptors (Figure 3C).

In addition to clearing damaged organelles by organellophagy, selective autophagy may eliminate invading pathogens, including bacteria and viruses, referred to as “xenophagy” [308,309,310,311,312]. Several types of cargo receptors, such as p62/SQSTM1, NDP52/Calcoco2, OPTN, and NBR1, participate in the process of xenophagy, in which *Salmonella* [313], group A *Streptococcus* [314], *Mycobacterium tuberculosis* [315], and Sindbis virus (SINV) [316] are eliminated. Phosphorylation of p62/SQSTM1 and OPTN [312,313,317] and ubiquitination of pathogen-associated proteins [318,319,320] play critical roles in the regulation of xenophagy (Figure 3C).

### 3.3. Virophagy

Several viral infections have been extensively shown to activate host autophagy to eliminate the infecting viruses and thus restrict viral growth in the infected cells, which is called “virophagy” [321,322,323]. The targeting of invading viruses for degradation through virophagy represents one type of cell defense against numerous human pathogenic viruses, including herpes simplex virus-1 (HSV-1) [324,325,326], SINV [327], human cytomegalovirus (HCMV) [328], Chikungunya virus (CHIKV) [329], and coxsackievirus B3 (CVB3) [330]. The RNA-dependent protein kinase (PKR) and eukaryotic translation initiation factor 2 subunit 1 (eIF2alpha)-dependent activation of virophagy degrades the HSV-1 proteins, thus restricting HSV-1 infection [324]. Conversely, ICP34.5 can antagonize HSV-1-induced virophagy by binding to Beclin-1 to inhibit autophagy initiation [331], thereby promoting HSV-1 replication and neurovirulence. Moreover, Ames et al. recently showed that OPTN promoted the degradation of HSV-1 VP16 and gB proteins through autophagy, and deficiency of OPTN led to impairment of host immunity, an increase in susceptibility to encephalitis, and loss of cognition [332]. SINV infection induces UBL-dependent virophagy to degrade viral capsid proteins through SMAD ubiquitin regulatory factor 1 (SMURF1)-mediated capsid ubiquitination and the recognition of ubiquitinated capsids by p62/SQSTM1 [316,327]. In a similar fashion, virophagy also promotes the degradation of the CHIKV capsid protein and CVB3 VP1 protein [329,330], indicating that virophagy may act as an innate antiviral immune response to negatively repress viral replication and protect infected cells from virus-induced cell death.

Since autophagy involves the rearrangement of intracellular membranes, several viruses exploit activated autophagy in infected cells to form multi-vesicular compartments for viral RNA replication, such as poliovirus and HRV [333,334]. Analogously, other viruses that cause a major burden in the human population, including HCV [335,336,337], hepatitis B virus (HBV) [338,339], DENV [340,341], and enteroviruses [342,343], were also shown to utilize autophagy in order to organize membranous structures for virus replication. Notably, SARS-CoV-2 infection has recently been reported to induce the accumulation of autophagosomes, in which viral RNAs replicate, presumably through the repression of autophagosome–lysosome fusion by the viral genome-encoded protein open reading frame 3a (ORF3a) [344,345,346]. Similarly, CVB3 infection has been shown to block autophagosome fusion through the viral protease-mediated proteolysis of SNARE complex, ultimately promoting accumulated autophagosome for the replication of the virus [347]. Therefore, viral-induced autophagy may play a distinct role in positively promoting virus growth by generating membranous vesicles for the replication of viral RNA. In addition, viral-induced autophagy functions in the nonlytic and intercellular spread of nonenveloped viruses, such as poliovirus, through an unconventional secretion pathway [348]. Moreover, autophagy reportedly participates in the viral envelopment of HBV [349], the egress of HCV [350], the maturation of infectious particles of DENV [351], and the prevention of cell death induced by DENV and CHIKV [352,353]. Together, these studies indicate that viral-induced autophagy may affect many aspects of the virus life cycle to benefit viral growth.

## 4. Regulation of RLR Signaling by Autophagy

In recent years, autophagy has emerged as playing a functional role(s) in negatively regulating RLR-mediated antiviral signaling. Several viruses activate autophagy and thus target the degradation of functional molecules and organelles involved in the regulation of the RLR IFN response. In this section, we therefore summarize the current knowledge on how autophagy represses RLR innate immunity according to the molecular targets of RLR signaling degraded by autophagy. The molecular mechanism and physiological significance of autophagy in RLR antiviral immunity is also discussed.

### 4.1. The Interaction between the ATG12-ATG5 Conjugate and CARDs of RIG-I and MDA5

The integration of autophagy into the RLR antiviral immune response was first uncovered by findings showing the enhancement of the vesicular stomatitis virus (VSV)-induced RLR-mediated IFN response in the infected ATG5-deficient mouse embryonic fibroblasts (MEFs) (Table 1) [354]. The increased RLR antiviral IFN immunity in VSV-infected cells lacking autophagy was characterized by increases in IFN-β mRNA levels and IRF3 phosphorylation, which were accompanied by a decrease in the viral infectivity of infected cells. Apart from VSV-infected cells, gene knockout of ATG5 also leads to hyperactivation of the dsRNA (poly[I:C])-induced type IFN I response, as shown by the increased mRNA levels of IFN-4α, IFN-β, interleukin-6 (IL-6), and C-X-C motif chemokine 10 (CXCL-10, also named IP-10). Reciprocally, overexpression of wild-type (WT) ATG5, rather than ATG5 K103R (Lys130 mutated to arginine [Arg]), which is unable to conjugate with ATG12 in cells, repressed the dsRNA-induced increase in promoter activities of IFN-4α, NF-kB, and IFN-β. In addition, the ATG12-ATG5 conjugate was shown to physically interact with RLRs in human embryonic kidney (HEK293) cells, including RIG-I and MAVS, through CARD binding. These studies collectively suggest that the ATG12-ATG5 conjugate, an important regulator of autophagosome maturation, may specifically bind to the CARDs of RIG-I and MAVS, leading to repression of RLR signaling and inhibition of the antiviral response (Table 1) (Figure 4) [354].

### 4.2. Removal of Mitochondria by Autophagy

Mitochondria and the associated ER membrane represent platforms for the assembly of RLR-associated molecules and signaling transduction of RLR antiviral immunity [119,120,121]. In recent years, it has been interesting and remains questionable whether alteration of mitochondrial dynamics and mitochondrial turnover regulate RLR antiviral signaling. Tal et al. firstly showed that interference with autophagy led to an increase in type I IFN response, which was associated with the accumulation of mitochondria [355]. In this study, the authors reported that gene silencing of ATG5 in MEF increased the dsRNA-induced production and secretion of IFN-α and IL-6, thus enhancing VSV infection (Table 1) [355]. In addition, ATG5-knockdown cells were shown to contain accumulated mitochondria, as demonstrated by the enhanced fluorescence intensity of MitoTracker (MitoTracker Green and MitoTracker Red)-labeled mitochondria in a flow cytometry assay and the upregulated mitochondrial DNA expression in a Southern blotting analysis. Along with these results, the expression of MAVS was indicated to be increased in ATG5 knockout MEFs, as shown by Western blotting and immunofluorescence (IFA) staining coupled flow cytometry assays. Additionally, overexpression of MAVS increased IFN-α mRNA levels in ATG5-deficient cells. Moreover, MitoSOX fluorescence probe labeling and flow cytometry experimental results showed that interruption of the autophagic process in MEF cells by ATG5 knockout led to an increase in the level of mitochondrial reactive oxygen species (ROS). In addition, treatment of ATG5-deficient MEFs with the antioxidant N-acetyl-L-cysteine (NAC) not only decreased the ROS level in mitochondria but also alleviated dsRNA-induced upregulation of IFN-α expression. In contrast, induction of mitochondrial ROS accumulation by an inhibitor of the mitochondrial electron transfer complex 1, rotenone, in autophagy-competent cells increased the dsRNA-activated IFN-α mRNA levels, which was further potentiated in rotenone-treated ATG5-deficient cells. In conclusion, these findings imply that autophagy plays a homeostatic role in the regulation of RLR antiviral immunity by eliminating mitochondria containing ROS, which may trigger overexpression of MAVS and hyperactivation of RLR signaling (Table 1) (Figure 4) [355].

### 4.3. Repression of Flaviviral PAMP-Triggered RLR Innate Immunity by Autophagy

In addition to VSV and dsRNA-induced IFN antiviral responses, viral-activated autophagy was also demonstrated to repress the HCV PAMP-induced IFN antiviral response (Table 1) [356,357]. The RIG-I N-terminal fragment (RIG-I N) has been previously shown to sufficiently activate the type I IFN response [22] and gene knockdown of ATG5 increased the RIG-I N-induced activation of IFN-β and ISRE promoters in human hepatoma Huh7 cells. Additionally, the HCV (JFH1 strain, genotype 2a) PAMP (3′-UTR and PU/UC within HCV RNA genome)-triggered IFN antiviral response was potentiated in cells with ATG5 gene silencing, as shown by the enhancement of IFN-β promoter activity and an increase in IFN-β mRNA level. The upregulated transactivation of IFN-β led to the upregulated expression of IFN-induced protein with tetratricopeptide repeats 1 (IFIT1, also named ISG56) and triggered paracrine antiviral immunity against HCV replication. In addition to HCV PAMPs, ATG5 knockdown also enhanced the DENV PAMP (3′-UTR)-induced activation of the IFN-β promoter and upregulated the expression of ISG56/IFIT1. Moreover, nutrient-starvation-induced autophagy triggered by treatment with Earle’s balanced salt solution (EBSS), Hank’s balanced salt solution (HBSS), or the mTOR inhibitor rapamycin was shown to repress HCV PAMP-activated IFN-β promoter activity. In a similar fashion, activation of autophagy through the unfolded protein response (UPR) by dithiothreitol (DTT) and tunicamycin treatment also inhibited the activation of the IFN-β promoter in HCV PAMP-treated cells. Furthermore, interference with autophagosome–lysosome fusion by the chemicals chloroquine (CQ) and bafilomycin (BAF-A1) and gene silencing of Rab7 and LAMP2 significantly inhibited IFN-β promoter activation. Overall, these results provide the first line of evidence showing that HCV activates autophagy to repress the RLR-mediated IFN response and suggest that pharmacological modulation of autophagy may alter innate antiviral immunity (Table 1) [356,357]. Soon after this study, another study showed that gene knockdown of Beclin-1 in HCV (H77 strain, genotype 1a)-infected immortalized human hepatocytes (IHHs) inhibited viral-activated autophagy and virus infectivity, which coincidently increased the mRNA levels of IFN-α, IFN-β, 2’-5’-oligoadenylate synthetase 1 (OAS1), and IFN-α-inducible protein 27 (IFI27) [358]. Additionally, inhibition of autophagy by silencing ATG7 gene expression analogously upregulated the mRNA expression of IFN-α, OAS1, and IFI27 and simultaneously reduced the infectious titer of HCV. Moreover, interference with HCV-induced autophagy promoted the apoptosis of infected IHHs. Again, these studies indicate that HCV may use autophagy to repress the RLR innate immune response (Table 1) [358]; however, whether and how RLR signaling molecules are regulated by autophagy in HCV-infected cells remain largely unclear.

### 4.4. Autophagic Degradation of TRAF6 by p62/SQSTM1

HCV (JFH1) infection was shown to induce autophagy through activating unfolded protein response (UPR) in an in vitro cell culture model [356,376]. Gene silencing of the molecules involved in autophagy and UPR inhibits the replication of HCV viral RNA [356,376], suggesting that HCV-activated autophagy benefits viral growth. In addition, HCV (JFH1) infection leads to mitochondrial fission and promotes the PINK1/Parkin-dependent mitophagy [377,378], thus attenuating cell apoptosis in the infected cells and promoting the establishment of viral persistence. These studies collectively indicate that HCV infection induces general autophagy and triggers selective autophagy to promote mitochondrial turnover.

Chan et al. first discovered that HCV (JFH1) infection led to the degradation of TRAF6, a ubiquitin E3 ligase necessary for producing polyubiquitin chains that recruit NEMO and activate RLR antiviral signaling (Table 1) [359]. Treatment with an autolysosome inhibitor, BAF-A1, restored the expression of TRAF6 in HCV-infected cells. Further studies revealed that HCV infection led to the colocalization of TRAF6 within viral-induced autophagic vacuoles and its interaction with p62/SQSTM1. Overexpression of TRAF6 inhibited the replication of HCV viral RNA, whereas depletion of TRAF6 gene expression increased the levels of HCV viral RNA and the viral infectivity of infected cells. Furthermore, TRAF6 knockdown was shown to reduce NF-kB promoter activity and decrease the mRNA and protein levels of IL-6 and tumor necrosis factor-α (TNF-α) in HCV-infected cells. These findings suggest that autophagy may negatively regulate the RLR-mediated type I IFN response by promoting the degradation of TRAF6 (Table 1) (Figure 4) [359].

### 4.5. Autophagic Degradation of RIG-I

RIG-I is a degradative substrate of autophagy induced by RNA viruses, including VSV and H1N1 IAV (Table 1) [360]. Du et al. reported that leucine-rich repeat containing protein 25 (LRRC25) significantly repressed RIG-I N-induced activation of the ISRE promoter in HEK293 cells (Table 1) [360]. The challenge of human leukemia monocytes, THP-1 cells with VSV and poly (I:C) resulted in the upregulation of LRRC25. Overexpression of LRRC25 repressed the poly (I:C)- and SeV-induced activation of the IFN-β and ISRE promoters, accompanied by enhanced VSV replication in HEK293 cells. In contrast, gene knockout of LRRC25 in THP-1 cells further increased VSV-triggered IRF3 phosphorylation and IFN-β production and simultaneously decreased the replication of VSV. Analogously, gene silencing of LRRC25 in human peripheral blood mononuclear cells (PBMCs) also led to the hyperactivation of H1N1 IAV-induced IRF3 phosphorylation and upregulation of IFN-β, IFIT1, and IFIT2 mRNA expression. In addition, VSV infection in THP-1 and PBMC cells and treatment with poly (I:C) in HEK293 cells promoted the interaction between LRRC25 and the CARDs of RIG-I and MDA5. Moreover, overexpression of LRRC25 in HEK293 cells promoted the degradation of RIG-I, which was abrogated by treatment with CQ and NH_4_Cl, inhibitors of autolysosome maturation and 3-MA, and gene knockout of ATG5 and Beclin-1. Moreover, the authors demonstrated that LRRC25 can mediate the interaction between RIG-I and p62/SQSTM1 in poly (I:C)-treated and VSV-infected cells by simultaneously binding to these two proteins. Gene knockout of p62/SQSTM1 reversed the LRRC25-induced degradation of RIG-I. It was noted that protein ubiquitination of RIG-I was not required for p62/SQSTM1-mediated targeting. Furthermore, gene knockout of ISG15 remarkably blocked the interaction between LRRC25 and RIG-I and reduced LRRC25-triggered RIG-I degradation, implying that ISG15 serves as a critical signal for the degradation of RIG-I by LRRC25. These studies collectively indicate that LRRC25 inhibits type I IFN response by promoting the degradation of RIG-I via ISG15 and p62/SQSTM1 (Table 1) [360]. In addition to LRRC25, the mitochondria-associated ubiquitin E3 ligase, membrane-associated ring-CH-type finger 5 (MARCH5), was also shown to promote the Lys-48-linked ubiquitination of RIG-I at the Lys193 and Lys203 residues within the CARDs of RIG-I, leading to the degradation of RIG-I and inhibition of RLR antiviral immunity (Table 1) (Figure 4) [361].

### 4.6. Degradation of RIG-I and MDA5 by Autophagy and PINK1/Parkin-Dependent Mitophagy

#### 4.6.1. Mitophagic Degradation of RIG-I and MDA5 by PINK/Parkin-Dependent Pathway

Induction of mitochondrial depolarization and mitophagy was recently reported to antagonize the type I IFN response (Table 1) [362]. Bu et al. reported that treatment of HEK293 cells with the mitochondrial uncoupler carbonyl cyanide 3-chlorophenylhydrazone (CCCP) activated mitophagy and simultaneously reduced SeV-triggered activation of the IFN-β, ISRE, and NF-kB promoters. Similarly, CCCP treatment also inhibited the increase in IFN-β, ISG56/IFIT1, and ISG15 mRNA levels in SeV-infected cells in a dose- and time-dependent manner. Notably, CCCP treatment led to a dramatic decrease in the protein levels of RIG-I, MDA5, and MAVS and the phosphorylation of TBK1 and IRF3 in cells infected with SeV. Ectopic expression of Parkin repressed the SeV-induced type I IFN response and promoted viral replication in infected cells, whereas gene silencing of Parkin potentiated RLR antiviral immunity in VSV-infected cells and repressed the proliferation of VSV. In addition, ectopic expression of PINK1 along with Parkin further repressed the IFN antiviral response in SeV-infected cells. The PINK1/Parkin-induced repression of the type I IFN response and degradation of RIG-I and MDA5 were reversed by BAF-A1 treatment. Moreover, CCCP also promoted the interaction between Parkin and RIG-I and MDA5 on mitochondria, thus promoting Lys-48-linked ubiquitination of RIG-I and MDA5. These studies suggest that PINK1/Parkin-dependent mitophagy may repress RLR IFN immunity by promoting the ubiquitination and degradation of RIG-I and MDA5 (Table 1) (Figure 4) [362]. Very recently, Glon et al. reported that in addition to CCCP, oligomycin/antimycin A (O/A), AMBRA1-ActA, and Epstein–Barr virus (EBV) BHRF1 protein also inhibited IFN antiviral response through inducing mitophagy [363], arguing again that mitophagy plays a negative role in regulating RLR IFN response (Table 1) (Figure 4).

#### 4.6.2. Autophagic Degradation of RIG-I and MDA5 by CCDC50

Apart from the known cargo receptors for selective autophagy, coiled-coil domain-containing protein 50 (CCDC50) was recently reported to function as a novel cargo receptor of selective autophagy for targeting RIG-I and MDA5 for degradation (Table 1) [364]. Gene silencing of CCDC50 in mouse BMDMs and BMDCs amplified the SeV-induced increase in IFN-β mRNA levels and production of IFN-β and increased the phosphorylation of TBK1 in SeV-infected cells. SeV infection led to increases in the mRNA and protein levels of CCDC50. In a similar fashion, gene knockout of CCDC50 in mice potentiated the SeV- and VSV-induced type 1 IFN response, restricted the viral replication of VSV, and protected VSV-infected mice from death. Overexpression of CCDC50 in HEK293 cells diminished the SeV-triggered activation of the IFN-β, ISRE, and NF-kB promoters, whereas gene knockout of CCDC50 increased the IFN-β mRNA and protein levels and repressed the production of infectious titers in VSV-infected cells. In addition, overexpression of CCDC50 led to the degradation of RIG-I and MDA5, while CCDC50 knockout prolonged the protein half-lives of RIG-I and MDA5. Moreover, SeV infection enhanced the interaction of CCDC50 with RIG-I and MDA5, and ectopic expression of CCDC50 in SeV-infected cells promoted the degradation of these two proteins, which was attenuated by 3-MA, CQ, and NH_4_Cl treatment. Additionally, SeV infection resulted in the interaction of CCDC50 with LC3B and p62/SQSTM1 and enhanced the colocalization between CCDC50 and autophagic vacuoles containing GFP-LC3B and p62/SQSTM1 puncta. Moreover, CCDC50 specifically bound to Lys-63-linked ubiquitinated RIG-I and MDA5 and interacted with LC3B, thereby targeting RIG-I and MDA5 for autophagy for degradation. These findings collectively suggest that CCDC50 may decrease the antiviral IFN response of RLR by targeting RIG-I and MDA5 for autophagic degradation (Table 1) (Figure 4) [364].

### 4.7. Autophagic Degradation of MAVS

#### 4.7.1. Autophagic Degradation of MAVS by NDP52/Calcoco2

In addition to RIG-I and MDA5, MAVS have been extensively shown to be degraded by autophagy in repressing RLR antiviral response. Jin et al. first reported that selective autophagy may negatively regulate RLR-triggered IFN antiviral immunity through Tetherin (Table 1) [365]. In this study, the authors demonstrated that ectopic expression of Tetherin promoted the viral replication of VSV and SeV in infected A549 cells (human adenocarcinoma alveolar basal epithelial cells), accompanied by repression of IFN-β activation. Overexpression of Tetherin also reduced the RIG-I N-triggered activation of IFN-β and ISRE promoters in large T antigen-immortalized HEK293 (HEK293T) cells, whereas silencing of Tetherin in HEK293T and A549 cells potentiated the poly (I:C) and 5′-triphosphate-RNA-induced activation of IFN-β and ISRE promoters and elevated the mRNA levels of IFN-β, ISG56/IFIT1, and ISG54 (also named IFIT2). Similarly, gene knockdown of Tetherin led to elevated IFN-β and ISG56/IFIT1 mRNA levels, enhanced phosphorylation of TBK1 and IRF3 in SeV- and H1N1 IAV-infected cells, and coincidentally attenuated the replication of these two viruses. Tetherin abrogated RIG-I N, MDA5 N (the N-terminal fragment of MDA5 has been previously shown to sufficiently activate the type I IFN response [22]), and MAVS-triggered inductions on IFN-β and ISRE promoters. Further studies showed the specific interaction between Tetherin and MAVS, which was enhanced by H1N1 IAV infection. Additionally, poly (I:C) treatment induced the colocalization of Tetherin with MAVS in mitochondria. Ectopic expression of Tetherin resulted in MAVS degradation, which was reversed by treating cells with 3-methyladenine (3-MA), an inhibitor of the class III (PI(3)K) complex, autolysosome inhibitors, CQ, BAF-A1, and ammonia chloride (NH_4_Cl), and gene silencing of ATG5 and Beclin-1. In addition, Tetherin promoted the physical interaction between MAVS and NDP52/Calcoco2. Downregulation of NDP52/Calcoco2 gene expression alleviated the Tetherin-induced degradation of MAVS, attenuated the Tetherin-mediated repression of SeV-triggered IFN-β, ISG56/IFIT1, and ISG54/IFIT2 mRNA levels, and repressed the Tetherin-triggered enhancement of virus replication in SeV-infected cells. Moreover, Tetherin promoted the Lys-27-linked ubiquitination of MAVS at the Lys7 residue through the ubiquitin E3 ligase membrane-associated ring finger (C3HC4) 8 (MARCH8). Interference with the ubiquitination of MAVS by gene knockout of MARCH8 and a mutation of the ubiquitination site of MAVS (K7R) protected MAVS from degradation by SeV infection and Tetherin, and amplified SeV-triggered RLR antiviral IFN immunity, including elevated phosphorylation of TBK1 and IRF3 and upregulated mRNA levels of IFN-β, ISG56/IFIT1, and ISG54/IFIT2. Collectively, these studies conclude that Tetherin functions as a negative regulator of the RLR downstream IFN response through MARCH8-mediated ubiquitination of MAVS and autophagic degradation of MAVS via interaction with NDP52/Calcoco2 (Table 1) (Figure 4) [365].

In addition to SeV and VSV, NDP52/Calcoco2 also targets MAVS for degradation to inhibit RLR antiviral signaling in CVB3-infected cells [330]. Mukherjee et al. first reported that CVB3 may attenuate the type I IFN response through 3Cpro protease-mediated cleavage of MAVS and Toll receptor domain-containing adaptor inducing interferon-beta (TRIF) [379]. In addition to MAVS, CVB3 3Cpro targets MDA5 for proteolysis and leads to inhibition of the activation of IFN antiviral immunity [380]. These studies suggest that CVB3 represses the type I IFN immune response through the viral protease-mediated proteolysis of MAVS, TRIF, and MDA5. Interestingly, CVB3 3Cpro was reported to cleave SNAP29 and PLEKHM1, two molecules that critically function in autophagosome–lysosome fusion, thus inhibiting autophagic flux in the infected cells and inducing the accumulation of autophagosome for virus replication [347]. Mohamud et al. showed that p62/SQSTM1 and NDP52/Calcoco2 differentially regulated virus proliferation in CVB3-infected HeLa cells (Table 1) [330]. Gene silencing of NDP52/Calcoco2 dramatically reduced CVB3 replication in the infected cells, whereas p62/SQSTM1 knockdown in CVB3-infected cells elevated viral replication. In addition, overexpression of p62/SQSTM1 reduced CVB3 replication in the infected cells, while ectopic expression of NDP52/Calcoco2 enhanced it in CVB3-infected cells. In addition, both NDP52/Calcoco2 and p62/SQSTM1 were demonstrated to interact with the CVB3 VP1 protein. Further analysis revealed that CVB3 infection induced Lys-48- and Lys-63-linked protein ubiquitination of VP1. In addition, gene knockdown of NDP52/Calcoco2, rather than p62/SQSTM1, in CVB3-infected cells led to an increase in MAVS protein levels and the phosphorylation of TBK1. CVB3 infection induced the cleavage of NDP52/Calcoco2 after Q139 treatment by viral protease 3C, and the free C-terminal fragment of NDP52/Calcoco2 was able to promote the degradation of MAVS and the viral replication of CVB3 and repress the activation of poly(I:C)-induced IFN-β production in a similar fashion as the full-length form. These findings suggest that CVB3 infection may mediate the degradation of MAVS through NDP52/Calcoco2, thereby attenuating antiviral signaling downstream of RLR (Table 1) (Figure 4) [330].

#### 4.7.2. Autophagic Degradation of MAVS by the RNF34-NDP52/Calcoco2 Axis

He et al. found that gene silencing of RNF34 in VSV-infected THP-1 cells remarkably potentiated the production of IFN-β and IL-6, accompanied by a decrease in the infectious titer (Table 1) [366]. Overexpression of WT RNF34 but not the E3 ligase-knockout mutant (H342A) of RNF34 significantly reduced the VSV-triggered activation of the IFN-β and NF-kB promoters. In addition, RNF34 overexpression repressed RIG-I N, MAVS, and poly (I:C)-triggered IFN-β activation. Reciprocally, knockdown of RNF34 gene expression elevated the phosphorylation of TBK1 and IRF3 and increased the expression of ISG54/IFIT2 and ISG56/IFIT1 in VSV-infected cells. RNF34 was shown to directly bind to MAVS, as indicated by yeast two-hybrid and glutathione S-transferase (GST) pull-down assays. VSV infection led to colocalization and interaction of RNF34 with MAVS. Additionally, RNF34 promoted Lys-27- and Lys-63-linked ubiquitination of MAVS at Lys311. Moreover, VSV infection led to the degradation of MAVS, which was reversed by NH_4_Cl treatment and gene silencing of RNF34. Furthermore, VSV infection also induced the interaction of MAVS with WT NDP52/Calcoco2, but not with the UBA mutant (D439R/C443K) of NDP52/Calcoco2. In addition, ectopic expression of WT NDP52/Calcoco2, but not the UBA mutant of NDP52/Calcoco2, repressed MAVS-induced activation of the IFN-β promoter. In contrast, gene knockdown of NDP52/Calcoco2 inhibited the VSV- and RNF34-induced degradation of MAVS. Furthermore, gene knockdown of RNF34 attenuated the VSV-induced degradation of the mitochondrial proteins TOMM20 and HSP60 and coincidently enhanced IRF3 phosphorylation. Conversely, ectopic expression of RNF34 facilitated the degradation of TOMM20 and HSP60, accompanied by a dramatic decrease in MAVS protein levels. Moreover, the accumulation of deformed mitochondria was found in the RNF34-silenced VSV-infected cells. Additionally, RNF34 was shown to induce mitophagy, as indicated by the increased acidic signal of an mt-Keima reporter, which is generally used for analyzing mitophagic processes [381,382]. These studies collectively suggest that RNF34 promotes the degradation of MAVS through NDP52/Calcoco2 and mitophagy (Table 1) (Figure 4) [366].

#### 4.7.3. Degradation of MAVS by IAV PB1-F2/TUFM Axis-Induced Mitophagy

Varga et al. first showed that IAV (H1N1) PB1-F2 protein repressed the RIG-I N and MAVS-induced type I IFN response, and IAV PB1-F2 colocalized with MAVS, which suggests that IAV PB1-F2 may interact with MAVS and interfere with MAVS downstream antiviral signaling (Table 1) [368]. Recently, Wang et al. reported that the IAV (H1N1 and H5N1 strains) PB1-F2 protein specifically activated autophagy through the formation of autolysosomes (Table 1) [367], as demonstrated by an increased level of LC3-II and enhanced RFP^+^/GFP^-^ fluorescence intensity of an mCherry-GFP-LC3 autophagy reporter, which is typically used to interpret autophagic flux [383]. Further analysis revealed the colocalization of IAV PB1-F2-induced autophagic vacuoles with mitochondria. The IAV PB1-F2 proteins also induced the degradation of mitochondrially encoded cytochrome C oxidase II (MTCO20) and translocase of the outer mitochondrial membrane complex subunit 20 (TOMM20). Moreover, the IAV PB1-F2 protein also increased the RFP^+^/GFP^+^ signal of the pmRFP-GFP-Mito mitophagy reporter. These results collectively suggest that the IAV PB1-F2 protein can activate mitophagy [367].

In addition, the authors demonstrated that the IAV PB1-F2 protein interacted with the Tu translation elongation factor, mitochondrial (TUFM) [367]. TUFM is a mitochondrial protein which has been shown to bind to NLR family member X1 (NLRX1) and the ATG12-ATG5-ATG16 trimeric complex (Table 1) [369]. TUFM was also demonstrated to activate autophagy and inhibit RLR downstream type I IFN response (Table 1) [369]. Downregulation of endogenous TUFM expression by small interference RNA (siRNA) gene knockdown and CRISPR/Cas9-sgRNA gene knockout inhibited IAV PB1-F2-induced mitophagy, as indicated by the restored protein levels of MTCO2 and TOMM20, and disturbed colocalization of autophagic vacuoles with mitochondria [367]. Moreover, overexpression of IAV PB1-F2 repressed MAVS-induced activation of the IFN-β promoter, whereas knockdown of TUFM reversed this inhibitory effect. The IAV PB1-F2-induced repression of the MAVS-activated IFN-β promoter was reversed by gene silencing of ATG5. Additionally, ectopic expression of IAV PB1-F2 resulted in the degradation of MAVS, which was restored by TUFM knockdown. Gene knockout of TUFM significantly increased the production of IFN in IAV-infected cells, accompanied by the repression of infected cell viral infectivity. Conversely, overexpression of TUFM benefited the viral growth of IAV-infected cells. Finally, the authors demonstrated that IAV PB1-F2 contained an LIR for binding to LC3B and served as a mitophagy receptor. Together, these results indicate that IAV infection may activate PB1-F2/TUFM axis-induced mitophagy to promote the degradation of MAVS, thus repressing RLR IFN immunity (Table 1) (Figure 4).

#### 4.7.4. Autophagic Degradation of MAVS by NBR1 and RNF5

Analogously, ectopic expression of the H7N9 PB1 protein was also shown to inhibit SeV infection and poly(I:C)-induced activation of the IFN-β promoter, ISRE, and NF-κB promoters in HEK293 cells (Table 1) [370]. Accordingly, the overexpression of PB1 potentiated the SeV- and poly(I:C)-triggered upregulation of IFN-β, IFN-stimulated gene 15 (ISG15), ISG56/IFIT1, regulated upon activation normal T-cell expressed and secreted factor (RANTES), and oligoadenylate synthetase-like protein (OASL) mRNA levels. PB1 also diminished the phosphorylation of IRF3, TBK1, and Ik-Bα and reduced RIG-I N, MDA5 N, and MAVS-triggered activation of the IFN-β promoter, suggesting that PB1 significantly inhibited RLR antiviral signaling. Coimmunoprecipitation (co-IP), in vitro GST pull-down, and IFA assays revealed that PB1 directly interacted with MAVS. In addition, PB1 promoted the degradation of MAVS, which was restored by treatment with CQ and BAF-A1, two autolysosome inhibitors, and gene knockout of ATG7. Reciprocally, activation of autophagy by EBSS further enhanced PB1-triggered MAVS degradation. In addition, PB1 promoted the autophagic degradation of MAVS through the induction of RNF5-mediated Lys27-linked ubiquitination of MAVS and the physical interaction with NBR1. Moreover, overexpression of NBR1 reduced RIG-I N-induced IFN-β promoter activation and enhanced the infectivity of H7N9 IAV-infected cells. In contrast, gene knockout of NBR1 increased the SeV-triggered activation of the IFN-β promoter and decreased the H7N9 IAV infectious titer of infected cells. Finally, ectopic expression of RNF5 increased the virus titer of H7N9 IAV-infected cells, whereas RNF5 knockout repressed the viral infectivity of H7N9 IAV-infected cells. These findings indicate that IAV PB1 induces K27-linked ubiquitination of MAVS by RNF5 and promotes the autophagic degradation of MAVS through NBR1, thereby repressing the antiviral response of RLR (Table 1) (Figure 4) [370].

#### 4.7.5. Autophagic Degradation of MAVS by p62/SQSTM1 and HFE

Liu et al. showed that H7N9 IAV infection in mice decreased the mRNA and protein levels of the hereditary hemochromatosis gene (HFE). The HFE protein has been shown to regulate iron metabolism by binding to the transferrin receptor (TFRC/TFR1) (Table 1) [371]. Gene knockout of HFE in H7N9 IAV-infected mice inhibited viral replication and protected the infected mice from H7N9 IAV-induced death. Notably, depletion of MAVS in H7N9 IAV-infected HFE knockout (HFE^−/−^) mice restored the infectious titer of infected cells. Further studies demonstrated that HFE gene knockout further potentiated the H7N9 IAV-induced mRNA and protein levels of IFN-β in infected mice, as well as the phosphorylation of TBK1 and STAT1 in H7N9 IAV-infected mouse bone marrow-derived macrophages (BMDMs), suggesting that H7N9 IAV infection induced HFE expression to repress the antiviral IFN response in RLR. In a similar fashion, depletion of HFE gene expression amplified the poly (I:C)-stimulated IFN-β mRNA and MAVS protein levels and VSV-induced TBK1, IRF3, and STAT1 phosphorylation. Ectopic expression of HFE led to a dramatic degradation of mitochondria- and peroxisome-associated MAVS in HEK293 cells. In contrast, HFE gene knockout resulted in the stabilization of MAVS in H7N9 IAV-infected mouse BMDMs, mouse bone marrow-derived dendritic cells (BMDCs), and MEFs. Moreover, HFE was shown to interact with and promote autophagic degradation of MAVS, which was reversed by treatment with CQ and gene silencing of ATG5 and ATG7. Furthermore, p62/SQSTM1 targeted MAVS for degradation by interacting with HFE, and gene knockout of p62/SQSTM1 diminished the HFE-induced degradation of MAVS and inhibited HFE-repressed MAVS-triggered activation of IFN-β and ISRE promoters. Overall, this study shows that HFE promotes autophagic degradation of MAVS by interacting with p62/SQSTM1 in IAV-infected cells, thus repressing downstream RLR antiviral immunity (Table 1) (Figure 4) [371].

### 4.8. Degradation of RIG-I and MAVS by CSFV-Activated Autophagy

Classical swine fever virus (CSFV), an enveloped and positive-stranded RNA virus, was shown to activate autophagy in infected swine kidney cells, PK-15 cells, and 3D4/2 porcine macrophage cells, as indicated by the upregulated level of LC3B-II and the increased number of GFP-LC3-labeled punctate (Table 1) [372]. Interference with the autophagic process by gene silencing of Beclin-1 or LC3B triggered cell apoptosis in CSFV-infected cells and increased the mRNA levels of IFN-α, IFN-β, and ISGs, TNF superfamily member 10 (TNFSF10), and tumor necrosis factor receptor superfamily member 6 (TNFRSF6, also known as Fas/CD95). In addition, deficiency of autophagy by gene knockdown of Beclin-1 or LC3B in CSFV-infected cells resulted in accumulation of ROS, upregulation of mitochondrial DNA, and increased protein levels of RIG-I and MAVS, which were reversed by NAC, an inhibitor of ROS release. In contrast, the induction of ROS by rotenone in CSFV-infected cells increased the level of RIG-I. Moreover, treatment with NAC reduced cell apoptosis and increased viral replication in CSFV-infected autophagy-deficient cells. In contrast, rotenone repressed the production of CSFV in autophagy-repressed cells. Thereafter, Xie et al. showed that CSFV activated autophagy through repression of mTOR and the calcium/calmodulin-dependent protein kinase 2 (CAMKK2/CaMKKβ)-protein kinase AMP-activated catalytic subunit alpha (PRKAA/AMPK) axis [373]. Moreover, the authors demonstrated that CSFV infection repressed the production of type I IFN through the interaction between Beclin-1 and MAVS. These findings indicate that CSFV may activate autophagy to remove ROS, thus preventing the infected cell from undergoing apoptosis and repressing the activation of the RLR antiviral immune response (Table 1) (Figure 4).

### 4.9. Degradation of IRF3 Protein Stability by TRIM21 and NDP52/Calcoco2

Wu et al. reported that SeV infection of A549 cells, THP-1 cells, and PBMCs led to the degradation of IRF3, which was restored by 3-MA treatment and gene knockout of ATG5 and Beclin-1 (Table 1) [374]. Additionally, SeV infection promoted the binding of IRF3 to NDP52/Calcoco2 and enhanced their colocalization. Gene knockout of NDP52/Calcoco2 resulted in the stabilization and dimerization of IRF3 in SeV-infected cells. In addition, SeV infection triggered the Lys-27-linked ubiquitination of IRF3 at K313 by TRIM21 ubiquitin E3 ligase. The ubiquitination of IRF3 by TRIM21 was shown to serve as a signal for NDP52/Calcoco2-mediated autophagic degradation in cells upon SeV infection. In contrast, the DUB enzyme proteasome 26S subunit, non-ATPase 14 (PSMD14), reversed the degradation of IRF3. Gene silencing and gene knockout of PSMD14 destabilized IRF3 in SeV-infected cells. Further analysis revealed that gene silencing of PSMD14 led to an increase in IRF3 protein ubiquitination in SeV-infected cells, and PSMD14 directly removed the conjugation of the polyubiquitin chain from K313 of IRF3, suggesting that PSMD14 is required for maintaining the stability of IRF3. Moreover, the reduction in VSV-induced ISRE promoter activation in PSMD14-knockout HEK293 cells, the decreased TBK1 and IRF3 phosphorylation by gene knockdown of PSMD14 in VSV-infected cells, the repression of VSV-triggered IFN-β production in PSMD14-silenced A549 cells, and the reduction in VSV-activated secretion of IFN-β in PSMD14-knockdown A549 cells together implied that PSMD14 expression is required for maintaining the basal level of IRF3 to induce the RLR IFN response. Gene knockout of NDP52/Calcoco2 and interference with the protein ubiquitination of IRF3 by a mutation on K313 (K313) diminished the PSMD14-enhanced SeV-infection-triggered IFN response. Hence, these findings suggest that autophagy regulates IRF3 stability and the antiviral IFN response of RLR (Table 1) (Figure 4) [374].

### 4.10. Autophagic Degradation of TBK1 through NEDD4 and NDP52/Calcoco2

In addition to IRF3, TBK1 was shown to be regulated by selective autophagy. Xie et al. recently reported that a ubiquitin E3 ligase, neural precursor cell-expressed developmentally downregulated gene 4 (NEDD4), was capable of attenuating the type I antiviral IFN response (Table 1) [375]. Ectopic expression of NEDD4 repressed SeV-, VSV-, and poly (I:C)-triggered activation of IFN-β and ISRE promoters in HEK293T cells. In particular, NEDD4 overexpression in SeV-infected cells diminished the upregulation of IFN-β, ISG56/IFIT1, and ISG54/IFIT2 mRNA levels and simultaneously enhanced viral replication in the infected cells. Additionally, ectopic expression of NEDD4 repressed the RIG-I N, MDA5, MAVS, and TBK1-induced activation of the IFN-β and ISRE promoters. In addition, NEDD4 was shown to bind to TBK1 physically, and SeV infection promoted the interaction and colocalization between these two proteins. Moreover, NEDD4 induced the Lys-27-linked ubiquitination of TBK1 at the K344 residue and promoted the recognition of TBK1 by NDP52/Calccoco2. Loss of NEDD4-triggered ubiquitination of TBK1 and gene knockout of NDP52/Calccoco2 resulted in resistance of TBK1 to degradation by NEDD4 in SeV-infected cells and relieved the repressive effect of NEDD4 on SeV-induced IFN-β, ISG56/IFIT1, and ISG54/IFIT2 mRNA levels. Together, these studies indicate a repressive role of the NEDD4/NDP52 axis in the regulation of RLR antiviral immunity by targeting TBK1 for degradation (Table 1) (Figure 4) [375].

## 5. Conclusions and Perspectives

In the past decade, multiple lines of evidence have implied that autophagy plays functional roles in the regulation of RLR antiviral signaling. Both general and selective autophagy may negatively control the activation of RLR innate immunity by targeting RLR signaling-related molecules for degradation, including RIG-I, MDA5, MAVS, IRF3, and TBK1. Additionally, several fundamental cellular events are involved in the autophagic degradation of these RLR signaling components: (1) the ubiquitin E3 ligases mediate the conjugations of different types of polyubiquitin chain linkages, such as Lys-48, Lys-63, and Lys-27-linked ubiquitination, to the degradative substrates; (2) the targeting of ubiquitinated molecules to degradation within the autophagic process by specific cargo receptors; and (3) the attenuation of degradation by DUB-mediated removal of protein ubiquitination. Although the molecular mechanisms underlying the regulation of RLR innate immunity by autophagy have been proposed, several fundamental questions remain. For instance, whether and how oragnellophagy-mediated turnover of damaged organelles, such as mitochondria and peroxisomes, mainly and/or sufficiently represses the antiviral signaling of RLR is still unclear. In addition, it remains questionable whether these identified molecular mechanisms control RLR innate immunity against virus infection by autophagy in a physiologically relevant context, and these mechanisms should be tested in viral infection experimental models using small animals with competent immune systems. Moreover, the spatial and temporal autophagic regulation of the RLR-mediated type I IFN response remains unresolved. Furthermore, numerous SNPs have been extensively identified in the genes encoding autophagic regulators, and most of these have been shown to be relevant to the development and pathogenesis of human diseases, such as cancer and Crohn’s disease [128,384]; however, the underlying molecular mechanism remains unclear. Most importantly, whether the SNPs and germline mutations of autophagy-related molecules involved in the repression of RLR antiviral signaling regulate the viral susceptibility and pathogenicity of infecting viruses remains largely unknown. Therefore, further investigations are urgently needed to comprehensively understand the physiological importance of the crosstalk between autophagy and RLR signaling in the balance of host cell-virus interactions and the pathogenesis of human diseases.

## Figures and Tables

**Figure 1 cells-12-00956-f001:**
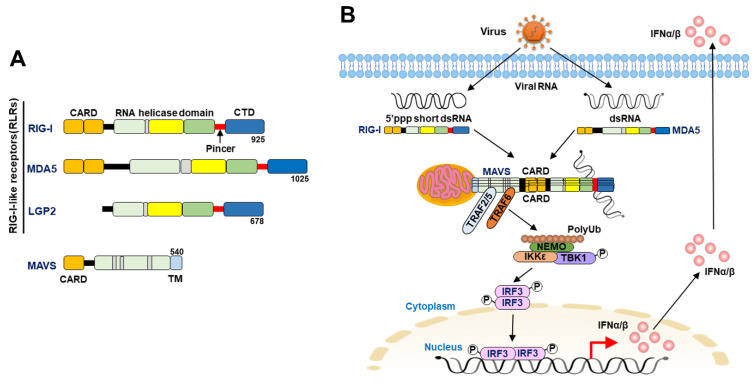
Signaling pathway of retinoic acid-inducible I (RIG-I)-like receptor (RLR) antiviral immunity: (**A**) Domain structure of RLRs, including retinoic acid-inducible I (RIG-I), melanoma differentiation association gene 5 (MDA5), laboratory of genetics and physiology 2 (LGP2), and the downstream effector mitochondrial antiviral signaling (MAVS). Among them, RIG-I, MDA5, and LGP2 contain the central helicase domain and C-terminal domain (CTD). There are two caspase activation and recruitment domains (CARDs) located within the N-termini of RIG-I and MDA5. MAVS has one N-terminal CARD that is able to associate with the CARDs of RIG-I and MDA5 and a C-terminal transmembrane domain for targeting to the mitochondrial outer membrane. (**B**) Activation of RLR antiviral signaling. During virus infection, the binding of viral RNAs induces conformational changes in RIG-I and MDA5. The free CARDs of RIG-I and MDA5 undergo self-oligomerization and interact with the CARDs of MAVS on mitochondria. The formed MAVS aggregates in turn to recruit the ubiquitin E3 ligases (TNF) receptor-associated factor 2 (TRAF2), TRAF5, and TRAF6, which produce polyubiquitin chains. The polyubiquitin chain then induces the recruitment of nuclear factor kappa-light-chain-enhancer of activated B (NF-kB) essential modulator (NEMO), which subsequently activates TANK binding kinase 1 (TBK1) to phosphorylate interferon response factor 3 (IRF3). Phosphorylated IRF3 then translocates into the nucleus and subsequently activates the gene expression of type I interferons (IFNs).

**Figure 2 cells-12-00956-f002:**
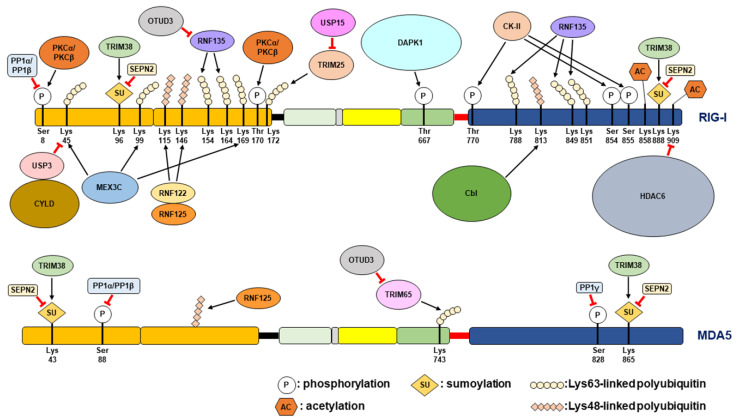
Post-translational modifications (PTMs) of RLR signaling molecules: RIG-I and MDA5 are regulated by phosphorylation, dephosphorylation, acetylation, deacetylation, ubiquitination, and deubiquitination and modified by Lys-48- and Lys-63-linked polyubiquitin. The enzymes responsible for PTMs are illustrated, and the modified amino acid residues are specifically indicated. PKC-α, Protein kinase C-α; PKC-β, Protein kinase C-β; TRIM25, tripartite motif 25; TRIM38, tripartite motif 25; TRIM65, tripartite motif 65; CK-II, casein kinase II; DAPK1, death-associated protein kinase 1; PP1α, protein phosphatase 1α; PP1-γ, protein phosphatase 1γ; HDAC6, histone deacetylase 6; SENP2, sentrin/SUMO-specific protease 2; RNF122, RING finger protein 122; RNF125, RING finger protein 125; RNF135, RING finger protein 135; MEX3C, mex-3 RNA binding family member C; Cbl, casitas B-lineage lymphoma proto-oncogene; USP4, ubiquitin-specific protease 4; USP15, ubiquitin-specific protease 15; CYLD, CYLD lysine 63 deubiquitinase; OTUD3, ovarian tumor protease deubiquitinating enzyme 3.

**Figure 3 cells-12-00956-f003:**
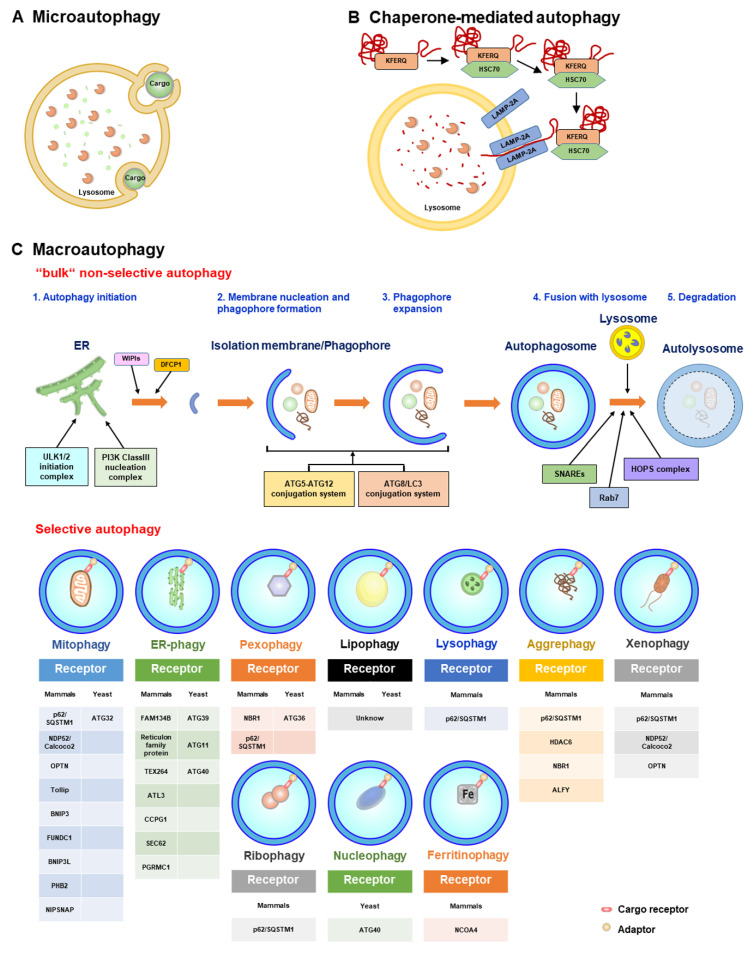
Three modes of autophagy and regulatory processes: (**A**) Microautophagy involves the invagination and/or protrusion of the lysosomal membranous structure for the engulfment of cytosol materials, including organelles, and the transfer of these materials to the lysosome lumen for degradation. (**B**) CMA recognizes the cargoes containing a KFERQ (Lys-Phe-Glu-Arg-Gln) motif by heat shock cognate 70 kDa protein (HSC70). Through docking with lysosomal membrane protein 2A (LAMP2A) on the membrane of lysosomes, the cargoes are delivered to the lysosomal lumen for degradation. (**C**) Macroautophagy is a vacuole biogenesis process consisting of membrane rearrangement events. The signaling cascade involved in the nutrient-starvation-induced mammalian target of rapamycin (mTOR) inhibition recruits the unc-51-like kinase 1/2 ULK1/2 initiation complex, which in turn activates the class III phosphatidylinositol-3-OH kinase (PI(3)K) complex for the biosynthesis of phosphatidylinositol-3-phosphate (PtdIns(3)P). The produced PtdIns(3)P subsequently recruits double-FYVE-containing protein 1 (DFCP1) and WD-repeat domain PtdIns(3)P-interacting (WIPI, also known as ATG18) to induce emergence of the IM/phagophore from the ER-associated membranous compartment. UBL conjugation systems (ATG5-ATG12 conjugation and ATG8/LC3 conjugation systems) facilitate the expansion and transformation of IM/phagophores into autophagosomes. Subsequently, autophagosomes fuse with lysosomes to form autolysosomes via SNAREs, Ras-related protein 7 (Rab7), and the homotypic fusion and protein sorting (HOPS) complex. The acidic lysosomal proteases degrade the sequestrated materials within autolysosomes. Selective autophagy specifically eliminates various types of organelles (including mitophagy, ER-phagy, pexophagy, ribophagy, lysophagy, lipophagy, and nucleophagy), aggregated proteins (aggrephagy), invading pathogens (xenophagy), and ferritin (ferritinophagy) via the recognition of cargo receptors. The interaction between the cargo receptor and ATG8/LC3 family protein selectively targets the cargo to the autophagic process for degradation. The cargo receptors for each type of selective autophagy in yeast and mammals are shown. p62/SQSTM1, p62/sequestosome 1; NDP52/Calcoco2, calcium-binding and coiled-coil domain-containing protein 2; NBR1, the neighbor of BRCA1; OPTN, optineurin; BNIP3, BCL2/adenovirus E1B 19 kDa protein-interacting protein 3; BNIP3L/Nix, BNIP3-like; Tax1 bp1, Tax1-binding protein 1; Tollip, Toll-interacting protein; ANT, adenine nucleotide translocator; FUNDC1, FUN14 domain-containing 1; ATG32, autophagy-related gene 32; PHB2, prohibitin 2; NIPSNAP, nitrophenylphosphatase domain and nonneuronal SNAP25-like protein homolog; ATG11, autophagy-related gene 11; ATG39, autophagy-related gene 39; ATG40, autophagy-related gene 40; FAM134B, the family with sequence similarity 134; TEX264, testis-expressed sequence 264 protein; ATL3, Atlastins 3; CCPG1, cell-cycle progression gene 1; SEC62, preprotein translocation factor 62; PGRMC1, progesterone receptor membrane component 1; ATG36, autophagy-related gene 36; HDAC6, histone deacetylase 6; ALFY, autophagy linked FYVE domain protein; NCOA4, nuclear receptor coactivator 4.

**Figure 4 cells-12-00956-f004:**
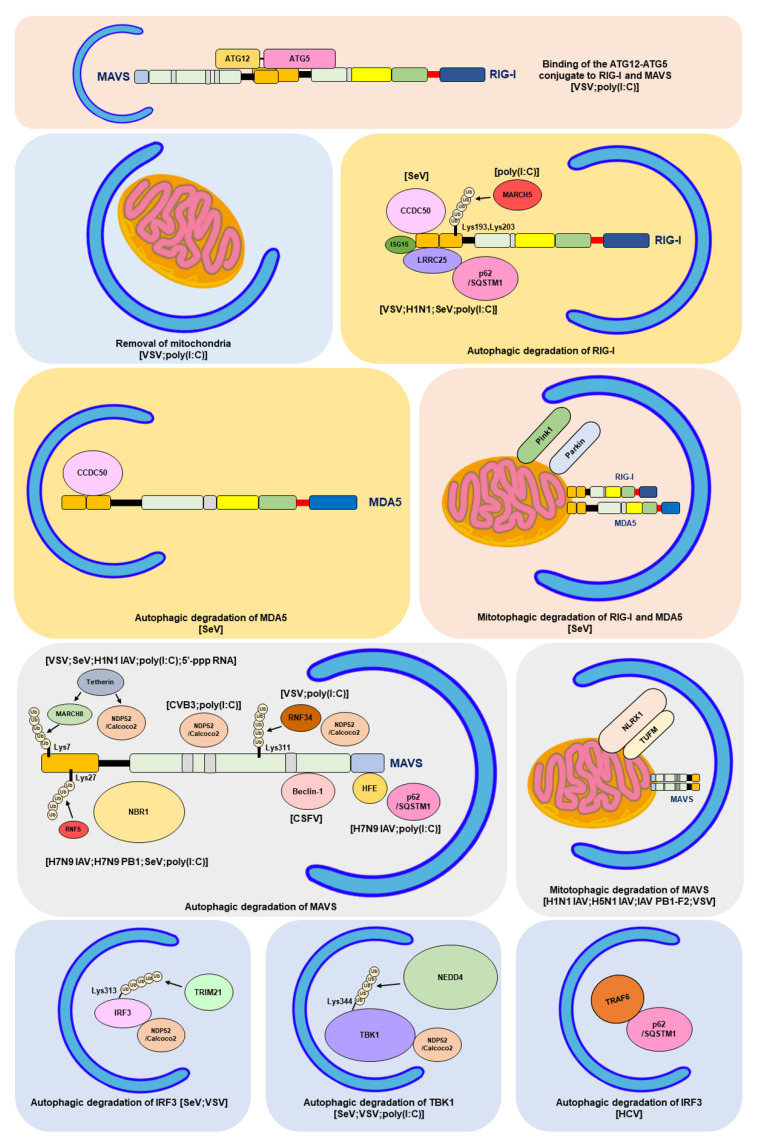
Summary of the negative regulation of RLR antiviral signaling by autophagy: The actions of autophagy-related molecules on mitochondria and the RLR signaling components RIG-I, MDA5, MAVS, TRAF6, IRF3, and TBK1 are shown. The cargo receptors, ubiquitin E3 ligases, and other factors involved in the degradation of RLR antiviral factors are illustrated. The amino acid residues of these molecules that could be ubiquitinated by ubiquitin E3 ligases are specifically indicated. poly(I:C), polyinosinic acid-polycytidylic acid; HCV, hepatitis C virus; VSV, vesicular stomatitis virus; SeV, Sendai virus; IAV, influenza A virus; CVB3, coxsackievirus B3; CSFV, Classical swine fever virus; LRRC25, leucine-rich repeat containing protein 25; MARCH5, membrane-associated ring-CH-type finger 5; ISG15, IFN-stimulated gene 15; CCDC50, coiled-coil domain-containing protein 50; PINK1, PTEN-induced putative kinase 1; RNF34, RING finger protein 34; MARCH8, membrane-associated ring finger (C3HC4) 8; RNF5, RING finger protein 5; TUFM, Tu translation elongation factor, mitochondrial; NLRX, NLR family member X1; TRM21, tripartite motif 21; HFE, hemochromatosis gene; NEDD4, neural precursor cell-expressed developmentally downregulated gene 4.

**Table 1 cells-12-00956-t001:** Negative regulation of RLR antiviral signaling by autophagy.

Target in RLR Signaling	Virus/PAMP RNAs	The Biological Function of Autophagy on RLR Molecules	Reference
RIG-I; MAVS	VSV; poly(I:C)	Binding of the ATG12-ATG5 conjugate to RIG-I and MAVS	[354]
Mitochondria	VSV; poly(I:C)	Elimination of mitochondria through autophagy	[355]
Unknown	HCV PAMP; DENV PAMP	Repression of the RLR IFN response by autophagy	[356,357]
Unknown	HCV	Repression of the RLR innate immune response by autophagy	[358]
TRAF6	HCV	Autophagic degradation of TRAF6 by p62/SQSTM1	[359]
RIG-I	VSV; H1N1 IAV; SeV; poly (I:C)	Autophagic degradation of RIG-I through LRRC25 and p62/SQSTM1	[360]
RIG-I	poly(I:C)	Autophagic degradation of RIG-I by MARCH5	[361]
RIG-I; MDA5	SeV; EBV	Autophagic degradation RIG-I and MDA5 by PINK1/Parkin-dependent pathway	[362,363]
RIG-I; MDA5	SeV	Autophagic degradation of RIG-I and MDA5 by CCDC50	[364]
MAVS	VSV; SeV; H1N1 IAV; poly(I:C); 5′-triphosphate-RNA	Autophagic degradation of MAVS by NDP52/Calcoco2	[365]
MAVS	CVB3; poly(I:C)	Autophagic degradation of MAVS by NDP52/Calcoco2	[330]
MAVS; mitochondria	VSV; poly(I:C)	Removal of mitochondria and autophagic degradation of MAVS by RNF34-NDP52/Calcoco2 axis	[366]
MAVS in mitochondria	H1N1 IAV; H5N1 IAV; IAV PB1-F2; VSV	Induction of mitophagy by IAV PB1-F2/TUFM axis and autophagic degradation of MAVS	[367,368,369]
MAVS	IAV H7N9; H7N9 PB1; SeV; poly(I:C)	Autophagic degradation of MAVS by NBR1 and RNF5	[370]
MAVS in mitochondria and peroxisome	H7N9 IAV; poly (I:C)	Autophagic degradation of MAVS by p62/SQSTM1 and HFE	[371]
RIG-I; MAVS	CSFV	Removal of mitochondria through autophagy and the interaction between Beclin-1 and MAVS	[372,373]
IRF3	SeV; VSV	Autophagic degradation of IRF3 by TRIM21-NDP52/Calcoco2 axis	[374]
TBK1	SeV; VSV; poly (I:C)	Autophagic degradation of TBK1 through NEDD4 and NDP52/Calcoco2	[375]

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
