# Peer review of "Crosstalk between Autophagy and RLR Signaling"

_cells, 2023, doi:10.3390/cells12060956_

Round 1
Reviewer 1 Report
Po-Yuan Ke wrote an up-to-date and interesting review on the interplay between autophagy and IFN induction via RLR signaling, in particular the role of autophagy in the inhibition of the RLR signaling.
The review begins with a long introduction on RLR antiviral signaling (with a very classical figure) and in this introduction a specific part on PTM of RIG-I and MDA5. This part should be supported by a figure because it is quite hard to follow. Then, autophagy is extensively described with a lot of general information and illustrated by a basic figure on autophagy without details, in particular regarding autophagy receptors, which are particularly described after. The part 4 of the article is about the regulation of RLR signaling pathway by autophagy and is exclusively on the negative regulation by autophagy, with a long list of 14 paragraphs describing each time in details one or two papers, rarely more. This is illustrated by just a table. The last paragraph is about the opposite, regulation of autophagy (activation) via RIG-I activation and is not fitting with the rest of the paper. The author needs to remove this in the text and in the table.
Globally, the review is interesting and up-to-date but the author should better put into perspective the different studies to obtain a synthesis of these mechanisms.
It will be really better if there were links between the different paragraphs of the part 4.
Specific remarks
This review will be greatly improved by adding figures in adequacy with the topic of the paper (or even replacing the figure 2). I suggest, in addition to the figure about PTM modifications mentioned above, one or several figures summarizing the different mechanisms and the identified actors (cellular and viral proteins when it is the case).
Figure 1
The author could add TRAF2/5/6 in the figure.
Page 4
“After entering the nucleus, this STAT1-STAT2 homodimer associates with IRF9”
Replace after by Before
Page 5
Hepatitis B virus is a DNA virus
Phosphorylation at serine residue 7 within the CARD of RIG-I is in fact Ser8.
Page 9
Legend of Figure 2: Double membraned autophagosomes, remove the d double-membrane
Line 6 The, the ATG12-ATG5 the first A is missing
Page 10
“Mitophagy is a specific kind of “organellophagy” that uniquely target harmful mitochondria”
This sentence is inaccurate. Mitophagy can be triggered even for normal mitochondria in specific contexts, in red blood cells for example or the paternal mitochondria after fecundation. Please remove harmful. The word organellophagy is quite unusual. Prefer selective autophagy
Second last line, ATG32 is not present in mammalian cells, only in yeast. The mammalian homolog is supposed to be BCL2L13.
4 lines above, "promote the constitution of an omegasome" instead of the Reconstitution
Page 11 in 3.3 Virophagy
First define Virophagy, in comparison with Xenophagy presented just above
eIF2alpha is misspelled two times. The activity of eIF2alpha is not involved in the inhibition of autophagy by ICP34.5 (only BECN1).
In general, autophagy is beneficial for viral multiplication by several mechanisms and not only for viral RNA replication, for example non-lytic release, envelopment, decrease of cell death. This sould be quickly discussed.
Replace ref 316 by Ames et al, (OPTN is a host intrinsic restriction factor against neuroinvasive HSV-1 infection) Nature Comm 2021 , because Alexander et al is not adapted to talk about autophagy as cell defense against HSV-1. This paper is a good example of Virophagy
Page 12.
It is important to add an introduction to this part 4 to better define the contours.
Table 1
Since every line of the table is about repression of RLR signaling this could be removed to simplify and to focus on the mechanisms, p62-mediated autophagic degradation of X for example.
In the column regulator of autophagy, sometimes indicated genes correspond to the ones KD or KO in the studies, but there are not specifically involved. so this should be removed.
Suppression and Abolishment mean total inhibition. And it is not the case (IFN response is not totally abolished) , so replace by inhibition or repression which are more adapted (in the table, in the text).
Page 14
The author needs to keep in mind that autophagy plays a homeostatic role in the control of RLR signaling in general with several cellular regulators which can be activated by viral infection or PAMPs. No absolute need of damaged or dysfunctional mitochondria.
page 15
4.4 titer is unclear and unnecessarily complicated
In part 4.5 gene silencing of Beclin-1 or LC3B (instead of AND) two times in this paragraph
4.6
Degradation of TRAF6 by autophagy is already selective autophagy. Please remove “in addition to general autophagy” (paragraphs 4.6 and 4.7)
Replace degradation of MAVS “by” NDP52 by “via interaction with”
4.7
A lot of very long sentences in this part. Please clarify and cut in two sentences each time. First the author should start by the study of Varga et al (ref349), then presents ref 348. They already observed an increase of autophagy in the first study.
PB1-F2 instead of PB1 F2. This very specific and important IAV protein could be presented.
Page 18
Titer is inaccurate. HFE is not induced by IAV
Similarly the last sentence of the paragraph 4.9 is inaccurate. IAV infection does not induce HFE expression but rather decreases HFE (the first sentence of the paragraph is correct)
Page 19 4.10
The last sentence of the paragraph is naccurate. The authors did not look wether LRRC25 was induced by IAV . They studied IAV infection in KO cells for LRRC25.
Page 19 4.11
Other studies regarding CVB3 have to be discussed. In particular regarding the inhibition of the autophagic flux and degradation of MAVS, TRIF and MDA5 by the viral protease, which in turn block IFN response, even if autophagy is not involved.
Page 21 4.15
A recent study (more recent than ref 360) reported that different mitophagy inducers block IFN response (CCCP but also Oligomycin/antimycin, Ambra-1 actA and a viral protein BHRF1). Glon et al, 2022, Plos Pathogens
4.16
Last sentence: Replace abrogate by decrease or inhibit
4.17
Replace autolysosome maturation by either autophagosome maturation or autolysosome formation
Please check RIG-I is not always correctly written (RIG-1 or RIGI)
Author Response
Dear Reviewer:
Thank you for giving me the opportunity to resubmit my manuscript “Crosstalk between Autophagy and RLR Signaling” to Cells (Manuscript ID: cells- 2252254). I appreciate the thoughtful and constructive comments provided by the reviewer. The content of this revised manuscript has been improved based on the reviewer’s comments, and I have incorporated additional parts, including Figure 2 (Posttranslational modifications of RLR signaling molecules) and Figure 4 (Summary of the negative regulation of RLR antiviral signaling by autophagy) as suggested by the reviewer. In addition, I have reconstructed the content of section 4 “Regulation of RLR signaling by autophagy” and Table 1 in the revised manuscript according to the reviewer’s suggestions. The changes are shown in the revised manuscript, and point-by-point responses to each comment are listed below.
Point 1: The review begins with a long introduction on RLR antiviral signaling (with a very classical figure) and in this introduction a specific part on PTM of RIG-I and MDA5. This part should be supported by a figure because it is quite hard to follow. Then, autophagy is extensively described with a lot of general information and illustrated by a basic figure on autophagy without details, in particular regarding autophagy receptors, which are particularly described after. The part 4 of the article is about the regulation of RLR signaling pathway by autophagy and is exclusively on the negative regulation by autophagy, with a long list of 14 paragraphs describing each time in details one or two papers, rarely more. This is illustrated by just a table. The last paragraph is about the opposite, regulation of autophagy (activation) via RIG-I activation and is not fitting with the rest of the paper. The author needs to remove this in the text and in the table.
Globally, the review is interesting and up-to-date but the author should better put into perspective the different studies to obtain a synthesis of these mechanisms.
It will be really better if there were links between the different paragraphs of the part 4.
Response 1: I thank the reviewer for the thoughtful comments on our manuscript and recognition of the merit of our manuscript. I have added one figure to illustrate the posttranslational modifications of RLR signaling components (Figure 2 in the revised manuscript) (please see Figure 2 on page 6 in the revised manuscript). In Figure 3, I have added the content of the complexes involved in each stage of macroautophagy and the responsible cargo receptors for each type of selective autophagy (please see Figure 3C on page 9 in the revised manuscript). The revised manuscript has removed the last part of section 4 in the original submission regarding the activation of autophagy via RLR. We have reorganized the content of section 4 “Regulation of RLR signaling by autophagy” (please see line 46 on page 12 to line 9 on page 24 in the revised manuscript). I divided the content of section 4 according to the regulated RLR signaling molecules, such as RIG-I, MDA5, …. etc., and set up several subtitles on each sub-section of section 4 (please see line 46 on page 12 to line 9 on page 24 in the revised manuscript). I also edited and consolidated the content of Table 1 (please see pages 13~14 in the revised manuscript).
Point 2: This review will be greatly improved by adding figures in adequacy with the topic of the paper (or even replacing the figure 2). I suggest, in addition to the figure about PTM modifications mentioned above, one or several figures summarizing the different mechanisms and the identified actors (cellular and viral proteins when it is the case).
Response 2: Thank you very much for the thoughtful suggestions. I have added Figure 2 in the revised manuscript, showing the posttranslational modifications of RLR signaling molecules (please see Figure 2 on page 6 in the revised manuscript). Also, Figure 4 showing “Summary of the negative regulation of RLR antiviral signaling by autophagy” was incorporated into the revised manuscript (please see Figure 4 on page 23 in the revised manuscript). In Figure 4 of the revised manuscript, the functional molecules involved in each aspect of autophagy-regulated RLR antiviral signaling are respectively illustrated.
Point 3: Figure 1, The author could add TRAF2/5/6 in the figure.
Response 3: I am very grateful for the reviewer’s thoughtful comment on Figure 1. I have incorporated the illustration on TRAF2, 5, and 6, and NEMO associated with RLR antiviral signaling in Figure 1 of the revised manuscript (Please see Figure 1 on page 3 in the revised manuscript).
Point 4: Page 4:“After entering the nucleus, this STAT1-STAT2 homodimer associates with IRF9”, Replace after by Before.
Response 4: I thank the reviewer for this comment. I have revised “After entering the nucleus…” to “Before entering the nucleus…” in the revised manuscript (Please see line 29 on page 3 in the revised manuscript).
Point 5: Page 5: Hepatitis B virus is a DNA virus. Phosphorylation at serine residue 7 within the CARD of RIG-I is in fact Ser8.
Response 5: I appreciate the reviewer for this comment. I have deleted the description of the hepatitis B virus (please see line 21 on page 4 in the revised manuscript). I have corrected “serine residue 7 within…” to serine residue 8 within …….” in the revised manuscript (Please see line 31 on page 4 in the revised manuscript).
Point 6: Page 9: Legend of Figure 2: Double membraned autophagosomes, remove the d double-membrane; Line 6 The, the ATG12-ATG5 the first A is missing.
Response 6: I thank the reviewer for this comment. I have removed “double-membrane” in the legend of Figure 3 in the revised manuscript (please see line 7 on page 10 in the revised manuscript). The “TG12-ATG5” has been corrected to “ATG12-ATG5” (please see line 11 on page 8 in the revised manuscript).
Point 7: Page 10: “Mitophagy is a specific kind of “organellophagy” that uniquely target harmful mitochondria”. This sentence is inaccurate. Mitophagy can be triggered even for normal mitochondria in specific contexts, in red blood cells for example or the paternal mitochondria after fecundation. Please remove harmful. The word organellophagy is quite unusual. Prefer selective autophagy. Second last line, ATG32 is not present in mammalian cells, only in yeast. The mammalian homolog is supposed to be BCL2L13. 4 lines above, "promote the constitution of an omegasome" instead of the Reconstitution.
Response 7: I am very grateful for the reviewer’s thoughtful comments. I have removed “harmful” in the sentence (please see line 52 on page 10 in the revised manuscript). I have revised “organellophagy” to “selective autophagy” in the sentence (please see line 52 on page 10 in the revised manuscript). I have fixed “ATG32” to “yeast ATG32” in the sentence (please see line 21 on page 11 in the revised manuscript). I have revised “Reconstitution” to “constitution” in the sentence (please see lines 16~17 on page 11 in the revised manuscript).
Point 8: Page 11: in 3.3 Virophagy, First define Virophagy, in comparison with Xenophagy presented just above. eIF2alpha is misspelled two times. The activity of eIF2alpha is not involved in the inhibition of autophagy by ICP34.5 (only BECN1). In general, autophagy is beneficial for viral multiplication by several mechanisms and not only for viral RNA replication, for example non-lytic release, envelopment, decrease of cell death. This sould be quickly discussed. Replace ref 316 by Ames et al, (OPTN is a host intrinsic restriction factor against neuroinvasive HSV-1 infection) Nature Comm 2021 , because Alexander et al is not adapted to talk about autophagy as cell defense against HSV-1. This paper is a good example of Virophagy.
Response 8: Thank you very much for the thoughtful suggestions. I have added the content that briefly introduces “virophagy” in section 3.3 (please see lines 2~7 on page 12 in the revised manuscript). The misspelling of eIF2alpha has been fixed (please see line 8 on page 12 in the revised manuscript). I have revised the content regarding the inhibition of autophagy by ICP34.4 (please see lines 10~12 on page 12 in the revised manuscript). I have incorporated the content for discussing the role of viral-activated autophagy in the non-lytic release, envelopment, decrease of cell death (please see lines 36~41 on page 12 in the revised manuscript). Reference 316 in the original submission has been removed, and the study reported by Ames et al. has been incorporated and discussed in the revised manuscript (please see lines 12~15 on page 12 in the revised manuscript).
Point 9: Page 12: It is important to add an introduction to this part 4 to better define the contours.
Response 9: I appreciate the reviewer’s thoughtful comments. An introduction to section 4 has been incorporated into the revised manuscript (please see lines 45~51 on page 12 in the revised manuscript).
Point 10: Table 1: Since every line of the table is about repression of RLR signaling this could be removed to simplify and to focus on the mechanisms, p62-mediated autophagic degradation of X for example. In the column regulator of autophagy, sometimes indicated genes correspond to the ones KD or KO in the studies, but there are not specifically involved. so this should be removed. Suppression and Abolishment mean total inhibition. And it is not the case (IFN response is not totally abolished), so replace by inhibition or repression which are more adapted (in the table, in the text).
Response 10: I thank the reviewer’s constructive suggestions. I have reconstructed the content of Table 1 in the revised manuscript (please see Table 1 on pages 13~14 in the revised manuscript). The “Physiological significance” has been revised to “The biological functions of autophagy on RLR molecules”. I used the simple description showing the action of autophagy regulators on RLR components, such autophagic degradation of RIG-I by LRRC25 and p62/SQSTM1, as suggested by the reviewer (please see Table 1 on pages 13~14 in the revised manuscript). The column showing the regulator of autophagy in the original submission has been removed (please see Table 1 on pages 13~14 in the revised manuscript). All the “Suppression” and “abolishment” words in Table 1 and the content of section 4 have been changed to “repression” and “inhibition” (please see Table 1 on pages 13~14 and section 4, please see line 46 on page 12 to line 9 on page 24 in the revised manuscript).
Point 11: Page 14: The author needs to keep in mind that autophagy plays a homeostatic role in the control of RLR signaling in general with several cellular regulators which can be activated by viral infection or PAMPs. No absolute need of damaged or dysfunctional mitochondria.
Response 11: I am very grateful for the reviewer’s constructive suggestion. I have removed the “damaged” and “dysfunctional” in the sentence of the revised manuscript (please see Table 1 on pages 13~14 and line 27 on page 14 in the revised manuscript).
Point 12: Page 15: 4.4 titer is unclear and unnecessarily complicated. In part 4.5 gene silencing of Beclin-1 or LC3B (instead of AND) two times in this paragraph.
Response 12: Thank you very much for the thoughtful suggestions. The title of section 4.4 in the original submission has been removed. I have revised the subtitles of section 4 according to the molecular target of RLR signaling regulated by autophagy, such as 4.4. Autophagic degradation of TRAF6 by p62/SQSTM1 in the revised manuscript (please see line 30 on page 15 in the revised manuscript). The “silencing of Beclin-1 and LC3B” has been fixed to “silencing of Beclin-1 or LC3B” in the revised manuscript (please see lines 8 and 12 on page 21 in the revised manuscript).
Point 13: 4.6: Degradation of TRAF6 by autophagy is already selective autophagy. Please remove “in addition to general autophagy” (paragraphs 4.6 and 4.7). Replace degradation of MAVS “by” NDP52 by “via interaction with”.
Response 13: I thank the reviewer’s constructive suggestions. I have removed the sentence “in addition to general autophagy” in section 4.7.1 in the revised manuscript (please see line 29 on page 17 in the revised manuscript) and section 4.7.3. (please see line 19 on page 19 in the revised manuscript). I have modified the sentence “by” to “via interaction with” in the revised manuscript (please see line 14 on page 18 in the revised manuscript).
Point 14: 4.7: A lot of very long sentences in this part. Please clarify and cut in two sentences each time. First the author should start by the study of Varga et al (ref349), then presents ref 348. They already observed an increase of autophagy in the first study. PB1-F2 instead of PB1 F2. This very specific and important IAV protein could be presented.
Response 14: I am very grateful for the reviewer’s constructive suggestion. I have reorganized the content of section 4.7. of the original submission, and the long sentences were split in section 4.7.3. the revised manuscript (please see lines 20~43 on page 19 in the revised manuscript). I have rearranged the content of section 4.7.3. in the revised manuscript (please see line 20 on page 19 to line 2 on page 20 in the revised manuscript). The study reported by Varga et al. was first introduced (please see lines 20~23 on page 19 in the revised manuscript). I have changed “PB1 F2” to “PB1-F2” in the sentences of section 4.7.3 (please see line 20 on page 19 to line 2 on page 20 in the revised manuscript).
Point 15: Page 18: Titer is inaccurate. HFE is not induced by IAV. Similarly, the last sentence of the paragraph 4.9 is inaccurate. IAV infection does not induce HFE expression but rather decreases HFE (the first sentence of the paragraph is correct).
Response 15: Thank you very much for this correction. I have removed the title of section 4.9. of the original submission and revised the title of section 4.7.5. to “Autophagic degradation of MAVS by p62/SQSTM1 and HFE” in the revised manuscript (please see line 28 on page 20 in the revised manuscript). The sentence described on “H7N9 IAV infection induces HFE expression…” of the original submission has been changed to “HFE promotes autophagic degradation of MAVS by interacting with p62/SQSTM1 in IAV-infected cells” (please see line 50 on page 20 to line 2 on page 21 in the revised manuscript).
Point 16: Page 19: 4.10 The last sentence of the paragraph is inaccurate. The authors did not look whether LRRC25 was induced by IAV. They studied IAV infection in KO cells for LRRC25.
Response 16: I am very grateful for your correction. The sentence “…H1N1 IAV infections induce LRRC25 expression….” has been deleted in the revised manuscript (please see lines 24~26 on page 16 in the revised manuscript).
Point 17: Page 19: 4.11 Other studies regarding CVB3 have to be discussed. In particular regarding the inhibition of the autophagic flux and degradation of MAVS, TRIF and MDA5 by the viral protease, which in turn block IFN response, even if autophagy is not involved.
Response 17: I appreciate the reviewer’s thoughtful suggestions. The studies regarding how CVB3 represses IFN response by protease have been incorporated into paragraph 2 of section 4.7.1 on page 18 in the revised manuscript (please see lines 18~24 on page 18 in the revised manuscript). The content of the study shown how CVB3 infection represses autophagic flux through the viral protease-mediated cleavage of SNAP29 and PLEKHM1 has been incoporated into the revised manuscript (please see lines 32~34 on page 12 and lines 24~27 on page 18 in the revised manuscript).
Point 18: Page 21: 4.15 A recent study (more recent than ref 360) reported that different mitophagy inducers block IFN response (CCCP but also Oligomycin/antimycin, Ambra-1 actA and a viral protein BHRF1). Glon et al, 2022, Plos Pathogens.
Response 18: Thank you very much for the constructive comments. The study reported by Glon et al. has been incorporated into section 4.6.1 in the revised manuscript (please see line 51 on page 16 to line 2 on page 17 in the revised manuscript).
Point 19: 4.16 Last sentence: Replace abrogate by decrease or inhibit
Response 19: I am very grateful to your correction. The word “abrogate” has been revised to “decrease” (please see line 26 on page 17 in the revised manuscript).
Point 20: 4.17 Replace autolysosome maturation by either autophagosome maturation or autolysosome formation.
Response 20: I appreciate the reviewer’s suggestion. As suggested by the reviewer in point 1. The content of section 4.17. is not fitted to the central theme in this review. I have removed this part in the revised manuscript (please see page 20 in the revised manuscript), and the sentence “autolysosome maturation” was no longer shown in the revised manuscript.
Point 21: Please check RIG-I is not always correctly written (RIG-1 or RIGI).
Response 21: Thank you for this comments. We have checked the spelling for RIG-I in the revised manuscript, and the RIG-I has been uniformly presented throughout the revised manuscript.
We hope that this version of our manuscript and our responses address all your concerns and that this revised manuscript meets the criteria for publication in Cells. Thank you for your kind consideration.
Sincerely,
Po-Yuan Ke, Ph.D.
Associate Professor
Department of Biochemistry & Molecular Biology and Graduate Institute of Biomedical Sciences, College of Medicine, Chang Gung University, Taoyuan 33302, Taiwan, Republic of China
Liver Research Center, Chang Gung Memorial Hospital, Linkou, Taoyuan 33305, Taiwan, Republic of China
Tel: 886-3-2118800-5115
E-mail: pyke0324@mail.cgu.edu.tw

Reviewer 2 Report
This timely review from Ke comprehensively discusses how different forms of autophagy are known to regulate RLR signaling as a strategy for blunting anti-viral immunity, describing molecular links between specific viral proteins, cellular autophagic mediators, and downstream protein or organelle targets. This is a fascinating subject worthy of further exploration, and the review does an excellent job of summarizing findings to date and highlighting gaps for future research. The review could be improved to some extent by (a) a more logical organizational strategy for section 4, and (b) some discussion of if/how inherited mutations in RLRs or autophagic proteins might disrupt homeostasis and contribute to heightened susceptibility to viral infections or even autoimmune conditions. The following critiques might help guide minor revisions to strengthen the review:
Major revisions:
· In general, section 4 would benefit from a more logical organizational structure, perhaps grouped by additional subheadings based on virus, autophagic target, or autophagic mediator.
· Pg 2: the discussion of PRR families seems to ignore NLRs/inflammasomes, and should be mentioned briefly
· Pg 5: There could be some discussion added of any known human mutations in RLR genes, autophagic machinery components, or their regulators that perturb anti-viral immunity. For example, germline mutations in MDA5 predispose individuals to severe/frequent infection with rhinoviruses, common coronaviruses and PIV.
· 4.3: How is HCV actually inducing autophagy? The included description in this section does not make it clear that HCV infection actually induces autophagy, although it is implied in next section (4.4) re: TRAF6 targeting. The author might consider moving this section first and/or explicitly stating that HCV induces autophagy, and describing evidence for whether it is bulk or selective macroautophagy.
Minor revisions:
· Pg 14: change RIG-1 to RIG-I
· Pg 16: change INF-b to IFN-b
Author Response
Dear Reviewer:
Thank you for giving me the opportunity to resubmit my manuscript “Crosstalk between Autophagy and RLR Signaling” to Cells (Manuscript ID: cells- 2252254). I appreciate the reviewer’s recognition of the merit of our manuscript and thank the reviewer for the thoughtful and constructive comments. The content of this manuscript has been improved based on the reviewer’s comments, and I have reconstructed the content of section 4 “Regulation of RLR signaling by autophagy” and Table 1 according to the reviewer’s comments. In addition, the discussion regarding how single nucleotide polymorphisms (SNPs) and germline mutations affect RLR antiviral signaling and autophagy-related molecules has been incorporated into the revised manuscript. The changes are shown in the revised manuscript, and point-by-point responses to each comment are listed below.
Point 1: In general, section 4 would benefit from a more logical organizational structure, perhaps grouped by additional subheadings based on virus, autophagic target, or autophagic mediator.
Response 1: Thank you very much for the thoughtful suggestions. As suggested by the reviewer, I have revised mainly the content of section 4 in the revised manuscript. The content of each sub-section in section 4 was categorized based on the autophagic target, such as RIG-I, MDA5, …. etc., and subtitles have been added in sub-section in the revised manuscript (please see line 44 on page 12 to line 9 on page 24 in the revised manuscript). The content of Table 1 of the revised manuscript has been consolidated and focused on the RLR signaling molecules regulated by autophagic regulators Table 1 (please see pages 13~14 in the revised manuscript). In Table 1 of the revised manusscript, the “Physiological significance” has been changed to “The biological functions of autophagy on RLR molecules”. I used the simple description showing the action of autophagy regulators on RLR components, such autophagic degradation of RIG-I by LRRC25 and p62/SQSTM1, as suggested by the reviewer (please see Table 1 on pages 13~14 in the revised manuscript). The column showing the regulator of autophagy in the original submission has been removed (please see Table 1 on pages 13~14 in the revised manuscript).
Point 2: Pg 2: the discussion of PRR families seems to ignore NLRs/inflammasomes, and should be mentioned briefly.
Response 2: I am very grateful for the reviewer’s comments. I have incorporated the content that briefly introduces NLRs/inflammasomes in section 2.1. in the revised manuscript (please see lines 16~20 on page 2 in the revised manuscript).
Point 3: Pg 5: There could be some discussion added of any known human mutations in RLR genes, autophagic machinery components, or their regulators that perturb anti-viral immunity. For example, germline mutations in MDA5 predispose individuals to severe/frequent infection with rhinoviruses, common coronaviruses and PIV.
Response 3: I appreciate the reviewer for the constructive suggestions. The content regarding how SNPs and germline mutations in RLR signaling molecules interfere with IFN antiviral response has been added in section 2.4. in the revised manuscript (please see line 26 on page 6 to line 13 on page 7 in the revised manuscript). The discussion on SNPs of autophagy regulators and their potential effects on RLRs have been incorporated into section 5 in the revised manuscript (please see lines 30~36 on page 24 in the revised manuscript).
Point 4: 4.3: How is HCV actually inducing autophagy? The included description in this section does not make it clear that HCV infection actually induces autophagy, although it is implied in next section (4.4) re: TRAF6 targeting. The author might consider moving this section first and/or explicitly stating that HCV induces autophagy, and describing evidence for whether it is bulk or selective macroautophagy.
Response 4: Thank you very much for the thoughtful comments. The content regarding how HCV induces autophagy and the introduction on the previous studies showing the activation of bulk and selective autophagy in the infected cells has been incorporated into section 4.4. in the revised manuscript (please see lines 31~38 on page 15 in the revised manuscript).
Point 5: Pg 14: change RIG-1 to RIG-I; Pg 16: change INF-b to IFN-b.
Response 5: I appreciate the reviewer for reminding and correction. The “RIG-1” has been revised to “RIG-I” in the revised manuscript (please see line 32 on page 14 in the revised manuscript). The “INF-b” has been revised to “IFN-b” in the revised manuscript (please see line 30 on page 17 to line 15 on page 18 in the revised manuscript).
We hope that this version of our manuscript and our responses address all your concerns and that this revised manuscript meets the criteria for publication in Cells. Thank you for your kind consideration.
Sincerely,
Po-Yuan Ke, Ph.D.
Associate Professor
Department of Biochemistry & Molecular Biology and Graduate Institute of Biomedical Sciences, College of Medicine, Chang Gung University, Taoyuan 33302, Taiwan, Republic of China
Liver Research Center, Chang Gung Memorial Hospital, Linkou, Taoyuan 33305, Taiwan, Republic of China
Tel: 886-3-2118800-5115
E-mail: pyke0324@mail.cgu.edu.tw
